# Inverse ceria-nickel catalyst for enhanced C−O bond hydrogenolysis of biomass and polyether

Zelun Zhao[1,3], Guang Gao[1,3], Yongjie Xi [1,3], Jia Wang[1], Peng Sun[1], Qi Liu[1], Chengyang Li[1], Zhiwei Huang[1] ✉ & Fuwei Li [1,2] ✉

Regulating interfacial electronic structure of oxide-metal composite catalyst for the selective transformation of biomass or plastic waste into high-value chemicals through specific C−O bond scission is still challenging due to the presence of multiple reducible bonds and low catalytic activity. Herein, we find that the inverse catalyst of $4CeO_x/Ni$ can efficiently transform various lignocellulose derivatives and polyether into the corresponding value-added hydroxyl-containing chemicals with activity enhancement (up to 36.5-fold increase in rate) compared to the conventional metal/oxide supported catalyst. In situ experiments and theoretical calculations reveal the electron-rich interfacial Ce and Ni species are responsible for the selective adsorption of C−O bond and efficient generation of $H^{\delta-}$ species, respectively, which synergistic facilitate cleavage of C−O bond and subsequent hydrogenation. This work advances the fundamental understanding of interfacial electronic interaction over inverse catalyst and provides a promising catalyst design strategy for efficient transformation of C−O bond.

The strong interface electronic interaction in a metal-oxide heterogeneous catalyst is essential in deciding its catalytic properties by tuning the adsorption and activation of adsorbed reactants and/or intermediates over the catalyst surface, where the distribution of interface electrons can be rearranged through charge transfer driven by the difference in the Fermi level of metal and oxide[1,2]. Interestingly, changing catalyst configuration from conventional supported metal catalyst over oxide support to the inverse oxide on metal has been employed as a promising tool for modulating the surface electronic state and found wide applications with enhanced catalytic activity and selectivity, particularly in the catalytic conversion of small molecules, such as $CO_2$, CO, and $O_2$[3–7]. Notably, the inverse $ZrO_2/Cu$ catalyst demonstrates 3.3 times higher activity compared to traditional $Cu/ZrO_2$ catalyst in the hydrogenation of $CO_2$ to methanol, since the strong electronic interaction induced interface Zr active sites on

inverse catalyst promote the adsorption of $CO_2$ and the subsequent formation of formate intermediates, which lead to the enhanced activity of $CO_2$ hydrogenation[3]. Moreover, the strong oxide-metal interaction facilitates the charge transfer from the metal Pd to oxide $RuO_x$ in the inverse $RuO_x/Pd$ catalyst and, therefore, changes the adsorption configuration of $O_2$ molecule, resulting in a 24-fold enhancement in the activity of oxygen reduction reaction compared to the Pd/C catalyst[4]. Nevertheless, the exploring application of inverse catalyst in the precise transformation of complex molecules and even polymer has been much less conducted, due to the challenges on selectively tuning the adsorption and activation of different functional groups[8–11].

Biomass-derived molecules and oxygen-containing polymer wastes have varied C−O bonds, such as etheric/furanic C−O bond in (hemi)cellulose derivates[12], phenolic C−O bond in lignin components[8],

[1]State Key Laboratory of Low Carbon Catalysis and Carbon Dioxide Utilization, State Key Laboratory for Oxo Synthesis and Selective Oxidation, Lanzhou Institute of Chemical Physics, Chinese Academy of Sciences, Lanzhou 730000, China. [2]School of Chemical Engineering, University of Chinese Academy of Sciences, Beijing 100049, China. [3]These authors contributed equally: Zelun Zhao, Guang Gao, Yongjie Xi. ✉e-mail: zwhuang@licp.cas.cn; fuweili@ucas.ac.cn

C(alkyl)−O bond in polyether[13], therefore, upgrading of biomass and recycling of plastic wastes through selective hydrogenolysis of specific C−O bond into value-added oxygenates with high atom-efficiency is a promising strategy towards carbon neutrality[14,15]. For example, ring-opening hydrogenation of tetrahydrofurfuryl alcohol (THFA) into 1,5−pentanediol (1,5−PDO) via selective cleavage of etheric C−O bond is a sustainable process for the synthesis of C5 diols compared to their production from petroleum feedstock at the industrial scale[16,17]. Although the development of noble metal (Ir, Pd, Ru, and Rh) catalysts efficiently improved the catalytic performance of the reaction by introducing $MO_x$ (M = Re, Mo, W, and V) in the last decades[18−23], it was found that the promotor $MO_x$ boosted the further hydrodeoxygenation of 1,5−PDO to the byproduct of 1−pentanol, resulting in that the reaction gave ideal selectivity to 1,5−PDO under low conversion and/or concentration of THFA[24−26]. Moreover, the high cost of catalysts and stability deterioration due to the leaching of $MO_x$ limit their application[16,27]. Unfortunately, these limitations become even more serious over recently developed Ni-based heterogeneous catalysts (Supplementary Table 1)[28−32]. Therefore, it is highly desirable to develop an efficient and long-lifetime non-precious metal catalyst for the hydrogenolysis of THFA, and hopefully for the other oxygenates containing etheric C−O bond as well.

Enlightened by effective tuning of strong electronic interaction on inverse oxide/metal catalyst and the resultant enhancement in catalytic performance, in this work, the ceria-nickel catalysts with inverse oxide/metal configuration were prepared by co-precipitation method. The inverse $4CeO_x$/Ni afforded a 98% 1,5−PDO selectivity under 95% conversion in the ring-opening hydrogenation of THFA, and demonstrated a promising catalytic activity of 36.5-fold in the reactive rate compared with conventional $96CeO_x$/Ni catalyst. Moreover, this inverse catalyst exhibited exceptional stability during a 200-hour continuous reaction, even in the absence of any solvent beyond the general requirement of a low concentration of reactant in the reported literature. The structure analysis of catalysts revealed the electronic state of the interfacial Ce and Ni species could be modulated by building the inverse oxide/metal configuration due to the strong electronic interaction. Further, in situ infrared Fourier transform (FT-IR) experiments and density functional theory (DFT) calculation results disclosed both the selective adsorption of etheric oxygen in THFA at the interface Ce sites adjacent to oxygen vacancy and the enhanced formation of $H^{\delta-}$ species at the interface Ni sites contributed to the superior catalytic performance of C−O bond hydrogenolysis. Interestingly, the inverse $4CeO_x$/Ni catalyst also exhibited enhancement in different etheric C−O bonds hydrogenolysis of the other representative biomass derivates and polyether to the desired alcohols. This work provides a promising example of boosting the non-precious metal-catalyzed utilization of biomass and oxygen-containing polymer waste to valuable hydroxyl compounds by tuning the interfacial electronic structure over inverse catalysts beyond conventional supported metal-oxide catalysts.

## Results

### Development of efficient and stable inverse Ce-Ni catalyst for THFA hydrogenation to 1,5−PDO

A coprecipitation method was employed to prepare a series of supported Ce-Ni catalysts[3], which were denoted as $nCeO_x$/Ni (n = 1, 4, 85, and 96) according to the Ce loadings determined by inductively coupled plasma mass spectrometry (ICP-MS) as shown in Supplementary Table 2. Using X-ray powder diffraction (XRD), the remarkable Ni crystal phase (PDF No: 65−2865) was observed in the $4CeO_x$/Ni and $1CeO_x$/Ni catalysts (Supplementary Fig. 1), and the negligible $CeO_2$ diffraction peaks could be ascribed to both the low content of Ce and the high dispersion of $CeO_x$ cluster[33]. In contrast, the strong diffraction peaks observed in the $96CeO_x$/Ni and $85CeO_x$/Ni catalysts indicated the presence of the characteristic fluorite $CeO_2$ phase (PDF No:

65−5923). Their great contrast in compositions suggested the different structural configurations of Ni and $CeO_2$ in the $nCeO_x$/Ni catalysts, whose geometrical structure was studied by high-resolution transmission electron microscopy, high-angle annular dark field scanning transmission electron microscopy, and energy-dispersive X-ray spectroscopy. As shown in Supplementary Fig. 2a, a representative inverse oxide/metal configuration of well-dispersed nano $CeO_x$ clusters (size of ~3 nm) supported on the Ni surface was observed over the $4CeO_x$/Ni catalyst. In contrast, the $96CeO_x$/Ni catalyst has a typical metal/oxide configuration, where the Ni particles (size of ~6 nm) were supported on the bulk $CeO_2$ (Supplementary Fig. 2b). Their geometrical structures were further studied by using element analysis through in situ X-ray photoelectron spectroscopy (XPS) combined with $Ar^+$ ion sputtering treatment (Supplementary Fig. 3). The intensity of Ni peak at 852.3 eV for inverse $4CeO_x$/Ni catalyst increased from $2.0 \times 10^5$ to $4.2 \times 10^5$ counts after 100 s of sputtering. Simultaneously, the intensity of the Ce peak at 885.1 eV diminished from $1.6 \times 10^4$ to $0.7 \times 10^4$ counts. In contrast, after sputtering, the enhancement of Ce signals and the weakness of Ni signals were observed over conventional $96CeO_x$/Ni catalyst as the erosion depth increased. These results demonstrated that Ce species were scattered on bulk Ni surface of inverse $4CeO_x$/Ni catalyst, while Ni species were distributed on the $CeO_2$ surface of conventional $96CeO_x$/Ni catalyst.

To develop an efficient non-precious metal catalyst for the sustainable upgrading of oxygen-containing biomass and plastic wastes, the ring-opening hydrogenation of THFA, derived from hemicellulose-based furfural, into 1,5−PDO was initially selected as a probing reaction. The inverse $4CeO_x$/Ni catalyst presented the highest production rate of 29.2 $\mu mol_{1,5−PDO}$ $g_{cat}^{-1}$ $min^{-1}$ under controlled THFA conversion of 63% (1,5−PDO selectivity of 98%) in a continuous flow fixed-bed reactor, which was 36.5 times than that of 0.8 $\mu mol_{1,5−PDO}$ $g_{cat}^{-1}$ $min^{-1}$ obtained over conventional $96CeO_x$/Ni catalyst (Fig. 1a, Supplementary Table 3). The physical mixed catalyst (PM-$4CeO_2$ + Ni) with the same molar ratio of Ce and Ni as $4CeO_x$/Ni catalyst produced a much lower 1,5−PDO production rate of 3.8 $\mu mol_{1,5−PDO}$ $g_{cat}^{-1}$ $min^{-1}$. In addition, trace or no 1,5−PDO was obtained over bulk Ni or $CeO_2$ alone. These results indicated that the interfacial Ce-Ni sites over inverse $4CeO_x$/Ni catalysts is responsible for the hydrogenolysis of C−O bond, where the synergy between Ce and Ni species is highly essential in the ring-opening hydrogenation of THFA[34]. To provide more information about the relationship between C−O bond hydrogenolysis activity and interface Ce-Ni sites, the reaction rate was normalized by their interfacial perimeter length, surface area, Ce mass, and Ni mass (Supplementary Fig. 4), which generally demonstrated the representative inverse $4CeO_x$/Ni catalyst exhibited superior catalytic activity compared to the representative conventional $96CeO_x$/Ni catalyst[33,35]. The carbon monoxide temperature-programmed desorption combined with mass spectrometry (CO-TPD-MS) results (Supplementary Fig. 5) of $nCeO_x$/Ni catalysts indicated a remarkable difference in the surface metallic sites between inverse and conventional catalyst because of the negligible CO desorption for the $4CeO_x$/Ni catalysts, compared to the strong desorption peaks at 250 °C and 600 °C for the $96CeO_x$/Ni catalyst[36]. Moreover, the oxidation of CO into $CO_2$ by $CeO_x$ species occurred at a lower temperature of 200 °C over the $4CeO_x$/Ni catalysts, as compared to the temperature of 550 °C required over the $96CeO_x$/Ni catalyst, which could be ascribed to the weakened Ce−O bond in inverse catalyst[37]. Based on the above results, it could be preliminarily deduced that the difference in the geometrical and electronic structure of inverse and conventional interfacial Ce-Ni sites contributed to the superior catalytic activity of the inverse catalyst compared to its conventional counterpart in the THFA hydrogenolysis.

Due to the unideal hydrodeoxygenation of 1,5−PDO to 1−pentanol over the catalysts with promoters such as $ReO_x$, $MoO_x$, $WO_x$, and $VO_x$ (Supplementary Table 4), the controlled low conversion and using low concentration of substrate were necessary to achieve high selectivity

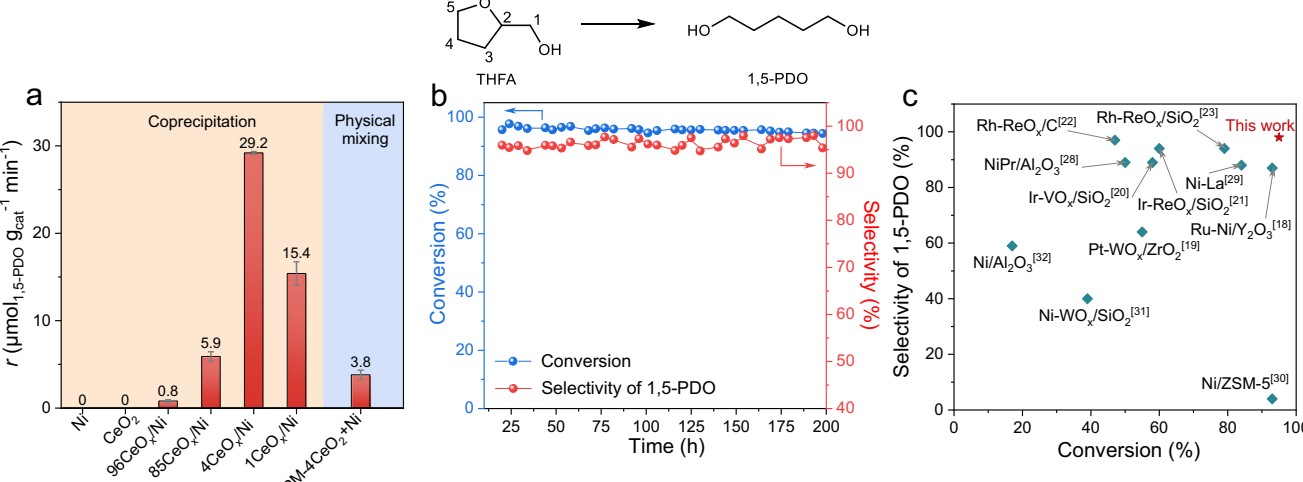

**Fig. 1 | Transformation of THFA into 1,5–PDO via etheric C−O bond hydrogenolysis. a** Catalytic performance over Ce-Ni catalysts. Reaction conditions: calcined catalyst: 5.4 g; THFA (solvent-free): 0.02 mL min⁻¹; 160 °C; 4.0 MPa H₂; gas-flow rate: 50 mL min⁻¹. Error bars refer to the standard deviation based on three measurements. **b** Catalyst stability experiment in a fixed-bed reactor over a 4CeOₓ/ Ni catalyst, calcined catalyst: 13 g, Red: selectivity of 1,5–PDO, blue: conversion of THFA. Before the reaction, the calcined catalyst was pre-reduced in an H₂/N₂ flow. **c** THFA hydrogenolysis over the representative supported catalysts[18–23,28–32]. THFA: tetrahydrofurfuryl alcohol, 1,5–PDO: 1,5–pentanediol.

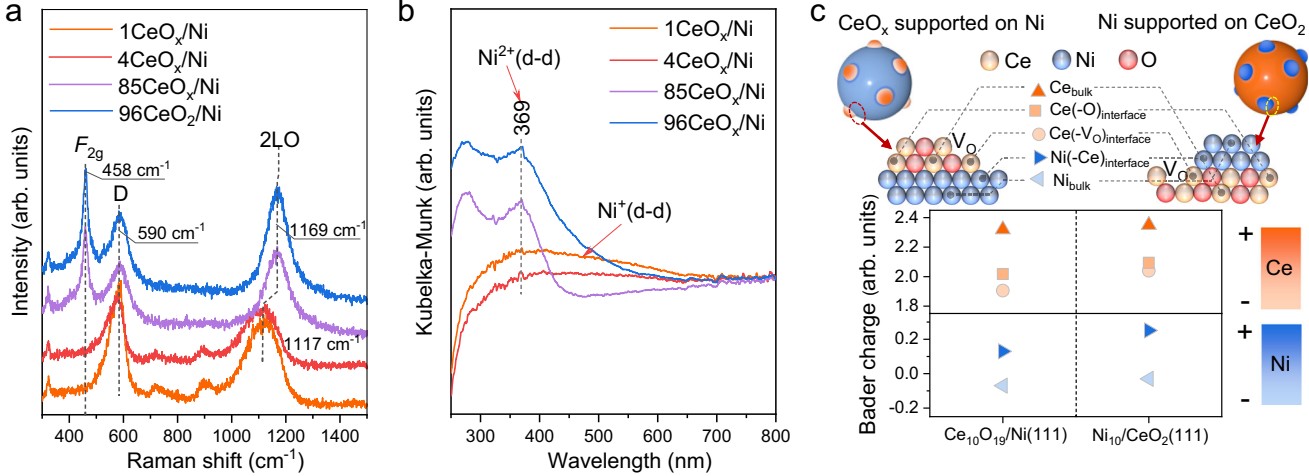

**Fig. 2 | The structural analysis of nCeOₓ/Ni catalysts. a** UV-Raman (325 nm) and **b** UV-vis adsorption spectra of 1CeOₓ/Ni (orange), 4CeOₓ/Ni (red), 85CeOₓ/Ni (purple), and 96CeOₓ/Ni (blue) catalysts. **c** Bader charge analysis of Ce and Ni species within inverse Ce₁₀O₁₉/Ni(111) and conventional Ni₁₀/CeO₂(111) catalysts, respectively.

of 1,5–PDO[24,25]. To our delight, under the high conversion of THFA above 95% and without using any solvent, 98% selectivity of 1,5–PDO was obtained over inverse 4CeOₓ/Ni catalyst (Fig. 1b). Those results are much better than the reported data obtained over even noble catalysts up to now (Fig. 1c and Supplementary Table 1)[18–23,28–32]. Notably, the inverse 4CeOₓ/Ni catalyst demonstrated good stability for 200 h in a fixed-bed reactor without remarkable decrease in the THFA conversion or 1,5–PDO selectivity at half and almost full conversion of THFA (Fig. 1b and Supplementary Fig. 6), which could be attributed to the inhibited agglomeration of metallic Ni species (Supplementary Fig. 7a)[38] and the enhanced stability in the Ce³⁺ species of CeOₓ cluster (Supplementary Fig. 7b)[5].

### Structural characterization of the CeOₓ/Ni catalysts
The valence state of surface Ce species was studied by quasi in situ XPS. As shown in Supplementary Fig. 8a, the 4CeOₓ/Ni and 1CeOₓ/Ni catalysts contained 67% and 70% of Ce³⁺ species by calculating the corresponding peak areas of Ce³⁺ and Ce⁴⁺ species[39], respectively,

which were more than those in the 96CeOₓ/Ni (41%) and 85CeOₓ/Ni (42%) catalysts, indicating that the reduction of Ce⁴⁺ to Ce³⁺ was promoted over the inverse catalyst. The chemical information of Ce species could also be determined by the ultraviolet Raman spectroscopy (UV-Raman, 325 nm). As shown in Fig. 2a, the $F_{2g}$ and D bands at around 458 and 590 cm⁻¹ observed over the conventional 96CeOₓ/Ni and 85CeOₓ/Ni catalysts were ascribed to the Ce–O stretch in CeO₂ lattice and oxygen vacancy related defect, respectively[40]. However, the negligible $F_{2g}$ bands for the inverse 4CeOₓ/Ni and 1CeOₓ/Ni catalysts revealed the increase in crystal structure disorder of the CeOₓ cluster on the inverse catalysts[40], which could be associated with the weakening of Ce–O bond, since the remarkable red-shift of the second order longitudinal optical mode (2LO) bands was observed from 1169 cm⁻¹ for the 96CeOₓ/Ni and 85CeOₓ/Ni catalysts to 1117 cm⁻¹ for the 4CeOₓ/Ni and 1CeOₓ/Ni catalysts[41].

Since the Ni⁶⁺ species was hard to identify by the Ni *2p* XPS spectra (Supplementary Fig. 8b)[42], the electronic state of surface Ni species was analyzed by ultraviolet-visible light (UV-vis) adsorption

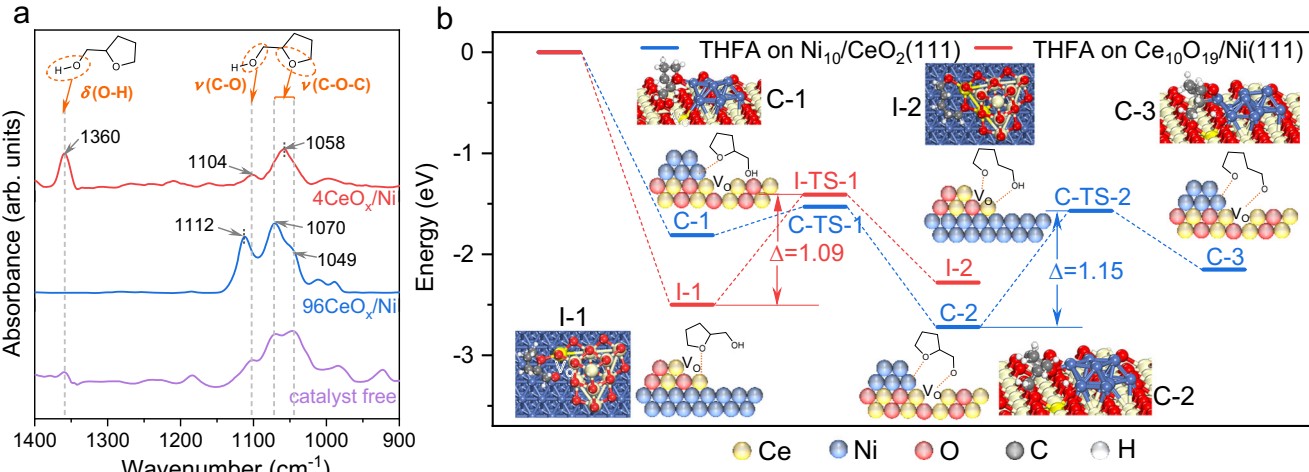

**Fig. 3 | Selective adsorption and activation of oxygenic groups on catalyst surface. a** In situ FT-IR spectra analysis of THFA (purple) and THFA adsorbed on 4CeO$_x$/Ni (red) and 96CeO$_x$/Ni (blue) catalysts. **b** DFT calculations for the selective adsorption of THFA and subsequent cleavage of etheric C2–O bond over Ce$_{10}$O$_{19}$/Ni(111) (red) and Ni$_{10}$/CeO$_2$(111) (blue), respectively. Atom colors: Ce (yellow), Ni (blue), C (gray), O (red), H (white).

spectroscopy. As shown in Fig. 2b, the strong adsorption bands at 369 nm are assigned to the d–d transition of Ni$^{2+}$ species in the 96CeO$_x$/Ni and 85CeO$_x$/Ni catalysts[43]. While, a new broadband from 400 to 600 nm appeared over 4CeO$_x$/Ni and 1CeO$_x$/Ni catalysts with a decrease in the intensity of Ni$^{2+}$ bands, which indicated the remarkable reduction of Ni$^{2+}$ to Ni$^{\delta+}$ (0 < $\delta$ < 1) species[42,44]. Hydrogen temperature-programmed reduction (H$_2$-TPR) experiments were performed to study the interaction between Ni and Ce species on the nCeO$_x$/Ni catalysts. As shown in Supplementary Fig. 8c, the peaks at 130 °C for the inverse 4CeO$_x$/Ni and 1CeO$_x$/Ni catalysts were attributed to the reduction of surface-adsorbed oxygen species associated with the formation of interfacial Ni–V$_O$–Ce[45]. In contrast, no remarkable reduction peak between 40 °C and 200 °C was observed over the conventional 85CeO$_x$/Ni and 96CeO$_x$/Ni catalysts, indicating that the reduction of interfacial Ni–O–Ce species over the conventional catalyst could occur at higher temperature, and the corresponding reduction peak might be overlapped with the broad peak at around 320 °C belonged to the reduction of NiO to metallic Ni[46]. The low reduction temperature of Ni–V$_O$–Ce species on the 4CeO$_x$/Ni and 1CeO$_x$/Ni catalysts indicated the strong interaction between interfacial Ni and Ce species on inverse catalyst[47], which could be further demonstrated by quasi in situ electron paramagnetic resonance (EPR) spectroscopy, where the *g*-value of Ni species shifted from 2.39 in the conventional 96CeO$_x$/Ni catalysts to 2.16 in the inverse 4CeO$_x$/Ni catalysts (Supplementary Fig. 8d)[42,48].

As indicated in the above catalyst screening tests, the interface catalytic sites containing Ni and Ce species were responsible for the enhanced hydrogenolysis of THFA in terms of catalytic activity and stability, particularly under solvent-free condition. Therefore, the structure of interface Ni-Ce was explored by using theoretical calculations with the models of Ce$_{10}$O$_{19}$/Ni(111) and Ni$_{10}$/CeO$_2$(111) representing the inverse and conventional catalyst, respectively[49,50]. Based on the difference in the coordination structure of Ce species, it could be categorized into three Ce species (Fig. 2c): Ce in the bulk CeO$_2$ (labeled as Ce$_{bulk}$), interfacial Ce without adjacent oxygen vacancy (labeled as Ce(–O)$_{interface}$), and interfacial Ce associated with an oxygen vacancy (labeled as Ce(–V$_O$)$_{interface}$). The corresponding Ni species were denoted as Ni$_{bulk}$ and Ni(–Ce)$_{interface}$, where the latter was defined as the interfacial Ni adjacent to the Ce. Bader charge analysis demonstrated that the presence of oxygen vacancy could modulate the electronic state[51], resulting in the lower valence state of Ce(–V$_O$)$_{interface}$ species (+2.04) than that of Ce(–O)$_{interface}$ species (+2.09) on the

conventional Ni$_{10}$/CeO$_2$(111) catalyst (Supplementary Fig. 9). It can be seen that the electronic interaction between the CeO$_x$ cluster and bulk Ni was significantly enhanced on the inverse Ce$_{10}$O$_{19}$/Ni(111) catalyst. Therefore, a larger electron density of Ce(–V$_O$)$_{interface}$ species (charge of +1.90) was generated. Simultaneously, the strong interaction also led to a lower Bader charge of Ni(–Ce)$_{interface}$ species on the inverse catalyst at +0.13 compared to that on the conventional catalyst at +0.25 (Supplementary Fig. 10). The results manifested that the electron transfer from the metallic Ni support to the interfacial Ce and Ni species was enhanced on the inverse oxide/metal catalyst due to the strong electronic interaction, which was supposed to enhance the catalytic activity of ring-opening hydrogenation of THFA into 1,5-PDO.

**Investigations on the selective adsorption of oxygenate groups on CeO$_x$/Ni calalysts**

Since the interfacial electronic effect could control the selective adsorption of substrate or intermediate[52], it is herein supposed that the strong electronic interaction over the inverse catalysts could tune the adsorption mode of THFA. After THFA saturated adsorption at 50 °C, a single adsorption peak at 1070 cm$^{-1}$ with a shoulder peak at 1049 cm$^{-1}$ for the 96CeO$_x$/Ni catalyst was detected using in situ FT-IR and was assignable to be the etheric C–O–C bond vibration referred to the corresponding adsorption peak of free THFA (Fig. 3a)[53]. While, a peak shift from 1070 cm$^{-1}$ to 1058 cm$^{-1}$ was observed over the 4CeO$_x$/Ni catalyst, indicating the difference in the adsorption mode of etheric C–O bond between the inverse catalyst and the conventional 96CeO$_x$/Ni catalyst. The adsorption peaks of C–O bond and O–H bond in the hydroxyl group on the 4CeO$_x$/Ni catalyst were located at 1104 cm$^{-1}$ and 1360 cm$^{-1}$, which are the same as those of free THFA[54,55]. However, when THFA was adsorbed on the 96CeO$_x$/Ni catalyst, the vibration peak of the O–H bond disappeared, revealing the cleavage of O–H bond, and simultaneously the adsorption peak of the hydroxyl C–O bond blue-shifted to 1112 cm$^{-1}$ suggesting the strong adsorption of the corresponding deprotonated hydroxyl oxygen which was supposed to locate at the oxygen vacancy sites on the conventional catalyst[56,57].

The selective adsorption of THFA was further studied by the DFT calculation. The THFA molecule was tried to adsorb at different interfacial sites on the inverse Ce$_{10}$O$_{19}$/Ni(111) catalyst (Supplementary Fig. 11), and it was found that THFA preferred to adsorb at the interface Ce sites due to the highest adsorption energy of −2.50 eV, where the etheric oxygen atom of THFA was bonded with the electron-rich Ce(–V$_O$)$_{interface}$ atom (I-1 in Fig. 3b and Supplementary Fig. 11a) rather

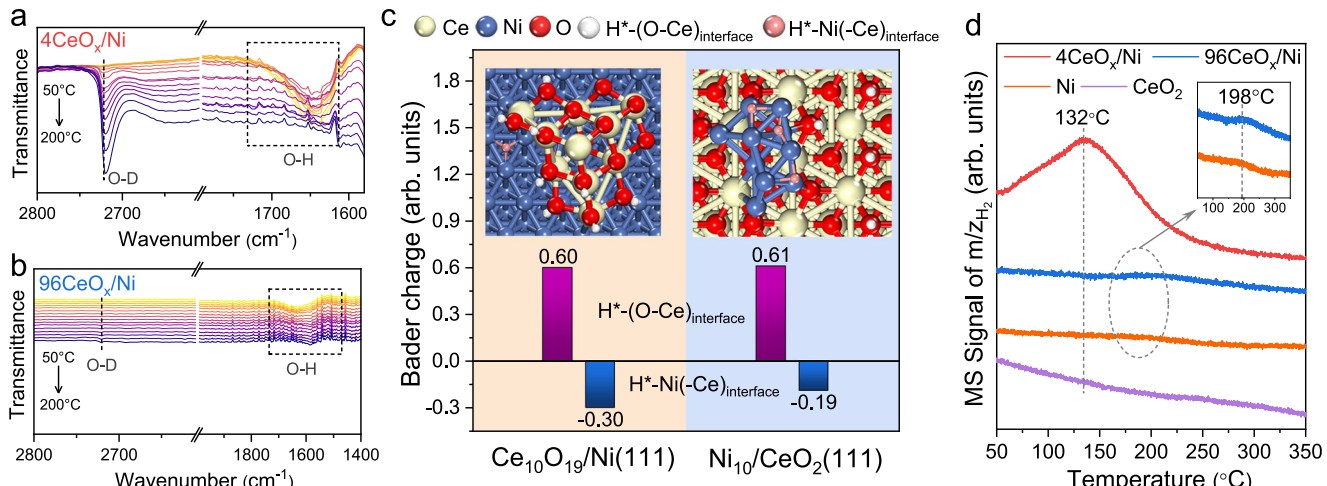

**Fig. 4 | Quantitative analysis of hydrogen species on catalyst surface.** The in situ FT-IR spectra for the analysis of surface hydrogen species in a $D_2$ flow: **a** inverse $4CeO_x/Ni$ catalyst and **b** conventional $96CeO_x/Ni$ catalyst. **c** Bader charge analysis of H species on $Ce_{10}O_{19}/Ni(111)$ and $Ni_{10}/CeO_2(111)$. Atom colors: Ce (yellow), Ni (blue), O (red), H (white and pink). **d** $H_2$-TPD-MS profiles over $4CeO_x/Ni$ (red line), $96CeO_x/Ni$ (blue line), Ni (orange line), and $CeO_2$ (purple line).

than $Ce(−O)_{interface}$ atom (Supplementary Fig. 11c). However, on the conventional $Ni_{10}/CeO_2(111)$ catalyst, the etheric oxygen atom adsorbed at the $Ni(−Ce)_{interface}$ site with a higher adsorption energy of −1.81 eV (C−1 in Fig. 3b and Supplementary Fig. 12a). The adsorbed THFA molecule on the $Ni_{10}/CeO_2(111)$ catalyst was unstable and proceeded a deprotonation process by overcoming a low barrier of 0.28 eV (C−1→C−2 in Fig. 3b and Supplementary Fig. 13), which afforded a stable intermediate through strong bonding with oxygen vacancy. The different adsorption modes of THFA could be further demonstrated by quasi in situ XPS adsorption experiment. The relative content of oxygen vacancy (determined by the ratio of $Ce^{3+}/(Ce^{3+}+Ce^{4+})$) and the surface-adsorbed O species associated with oxygen vacancy (denoted as $O_{II}$, calculated by its corresponding area ratio to lattice oxygen of $O_I$) were monitored, respectively[35,58]. As shown in Supplementary Fig. 14, the relative content of $O_{II}$ species increased from 1.6 to 2.9 with no remarkable change of oxygen vacancy after THFA adsorption over the inverse $4CeO_x/Ni$ catalyst. By contrast, the THFA adsorbed on conventional $96CeO_x/Ni$ catalyst led to the decrease in $O_{II}$ content from 0.6 to 0.5 with the diminishment of oxygen vacancy content ($Ce^{3+}/(Ce^{3+}+Ce^{4+})$ ratio from 41% to 34%). These results manifested the adsorption mode of THFA over inverse $4CeO_x/Ni$ catalyst via selective adsorption of etheric oxygen at the $Ce(−V_O)_{interface}$ site adjacent to oxygen vacancy, while both the adsorption of etheric oxygen at the $Ni(−Ce)_{interface}$ site and deprotonation of hydroxyl at the oxygen vacancy occurred simultaneously over conventional $96CeO_x/Ni$ catalyst.

The adsorption mode of THFA significantly affects the catalytic activity of etheric C−O bond cleavage which is the key step in the ring-opening hydrogenation of THFA. When the THFA was adsorbed on the $Ce_{10}O_{19}/Ni(111)$ catalyst, the adsorbed intermediate of I−1 underwent ring-opening via C2−O bond cleavage by overcoming a barrier of 1.09 eV (I−1→I−2 in Fig. 3b and Supplementary Fig. 15). However, a higher barrier of 1.15 eV was necessary for the cleavage of C2−O bond in the deprotonated intermediate of C−2 over the $Ni_{10}/CeO_2(111)$ catalyst (C−2→C−3 in Fig. 3b and Supplementary Fig. 13), which gave rise to the higher catalytic activity by inverse $4CeO_x/Ni$ catalyst than that of conventional $96CeO_x/Ni$ catalyst in the ring-opening hydrogenation of THFA into 1,5−PDO.

## Investigations on the adsorption and activation of hydrogen species on CeOx/Ni catalysts

The activated H species (i.e., H•, H⁻) via homolytic or heterolytic dissociation of $H_2$ have a profound influence on the catalytic

hydrogenation activity[59]. Before measurements by in situ FT-IR spectroscopy to study the formation of hydride species on the catalysts, the catalysts were pre-reduced by $H_2$. As shown in Fig. 4a, the O−H vibration bands between 1600−1720 cm⁻¹ were observed over the inverse $4CeO_x/Ni$ catalyst[60]. As the temperature increased from 50 °C to 200 °C under the $D_2$ atmosphere, the intensity of O−H peak gradually diminished, while the O−D stretching vibration peak appeared at 2720 cm⁻¹ when the temperature reached 100 °C, and the peak intensity increased with raising temperature, indicating the exchange reaction between $H^{δ+}$ and $D^{δ+}$ in surface hydroxyl group occurred on the inverse $4CeO_x/Ni$ catalyst[61]. However, no remarkable $H^{δ+}−D^{δ+}$ exchange reaction proceeded on the $96CeO_x/Ni$ conventional catalyst, since the negligible O−D vibration peak was observed in Fig. 4b. It was worth noting that both $D_2$ and $H_2$ could be readily activated and transformed into HD via H−D exchange reaction on the conventional $96CeO_x/Ni$ catalyst according to the obvious HD signal detected below 100 °C (Supplementary Fig. 16). The results manifested that H atom instead of $H^{δ+}$ species was formed on the conventional $96CeO_x/Ni$ catalyst via homolytic cleavage of $H_2$. In contrast, the formation of $H^{δ+}$ species suggested the heterolytic dissociation of $H_2$ on the inverse $4CeO_x/Ni$ catalyst, which simultaneously produced the corresponding $H^{δ−}$ species.

To study the location of hydrogen species on the inverse and conventional catalysts, the DFT calculations were conducted. As shown in Fig. 4c, the H species preferred to bond with the interfacial Ni atoms (denoted as $H^*−Ni(−Ce)_{interface}$) and O atoms of $CeO_x$ (denoted as $H^*−(O−Ce)_{interface}$) rather than with the interfacial Ce atom. Further, Bader charge analysis (Supplementary Fig. 17) shows that the valence state of $H^*−Ni(−Ce)_{interface}$ species (−0.30 a.u.) on $Ce_{10}O_{19}/Ni(111)$ was more negative than that (−0.19 a.u.) on $Ni_{10}/CeO_2(111)$, indicating that $H^{δ−}$ species tend to form on the inverse catalyst due to bonding with the electron-rich $Ni(−Ce)_{interface}$. Combining with the above structural analysis, the facile formation of $H^{δ−}$ species over inverse catalyst could be attributed to the strong electronic interaction which boosted the charge transfer from bulk Ni to the interfacial Ni.

It is recognized that the adsorbed H species can undergo recombination and desorb as $H_2$ molecule from the catalyst surface at an appropriate temperature. Here, an obvious $H_2$ desorption peak at 132 °C for inverse $4CeO_x/Ni$ catalyst was observed by hydrogen temperature-programmed desorption combined with mass spectrometry ($H_2$-TPD-MS, red line in Fig. 4d), which was attributed to the recombination between adsorbed $H^{δ+}$ and $H^{δ−}$ species[62]. While, the

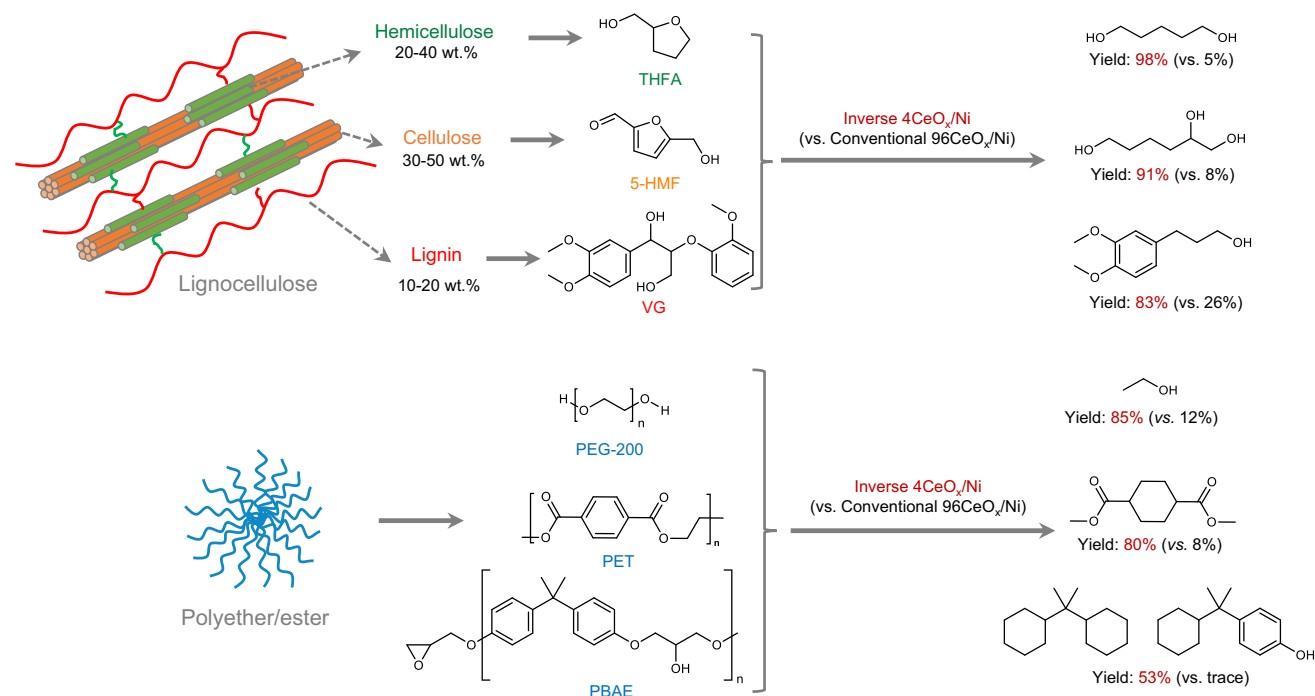

**Fig. 5 | Transformation of representative lignocellulose derivates and polyether/ester over inverse 4CeO_x/Ni catalyst.** THFA tetrahydrofurfuryl alcohol, 5–HMF 5–hydroxymethyl furfural, VG veratrylglycerol-β-guaiacyl ether, PEG polyethylene glycol, PET polyethylene terephthalate, PBAE poly(bisphenol A-co-epichlorohydrin) glycidyl end-capped.

$96CeO_x/Ni$ catalyst (blue line) desorbed $H_2$ at a higher temperature (198 °C) similar to that of metal Ni catalyst (orange line), indicating the desorption of atomic state H on the conventional catalyst[63]. Their relative content of adsorbed H species on $4CeO_x/Ni$ and $96CeO_x/Ni$ catalysts were calculated by area integrating of the corresponding peaks to be 9.64 and 0.77 $\mu mol_{H2}\ g_{cat}^{-1}$, respectively. The results were consistent with the great difference in their THFA hydrogenolysis activity as shown in Fig. 1a. The kinetic analysis of $H_2$ pressure further demonstrated the correlation of reactive activity with the content of H species, which could be tuned by changing $H_2$ pressure[24]. As shown in Supplementary Fig. 18, the specific rate of 1,5–PDO increased as the pressure increased from 0.4 MPa to 4.0 MPa. Further increase in $H_2$ pressure led to the saturated adsorption of H species on the $4CeO_x/Ni$ catalyst, and no significant improvement in catalytic activity over 4.0 MPa was observed. Therefore, the superior hydrogenolysis activity of the inverse catalyst could also be associated with the high content of $H^{\delta-}$ species on $4CeO_x/Ni$, since the $H^{\delta-}$ species was reported to prefer to reduce the polar group[64].

### Selective hydrogenation transformation of representative biomass-derived molecules and polyether

In addition to the ring-opening hydrogenolysis of THFA derived from hemicellulose (Fig. 5, Supplementary Table 5), the development of a general and efficient catalyst towards etheric C–O bond hydrogenolysis of other lignocellulose-derived molecules and even oxygen-containing plastic waste into valuable hydroxyl compounds was highly desirable and challenging. To our delight, 5–hydroxymethyl furfural (5–HMF), a platform compound readily obtained from cellulose[65], yielded 91% of 1,2,6–hexanetriol through the hydrogenolysis cleavage of furanic C–O bond on inverse $4CeO_x/Ni$ catalyst, while its conventional counterpart only gave 1,2,6–hexanetriol in 8% yield. For the selective cleavage of the C–O bonds of veratrylglycerol-β-guaiacyl ether (VG), a representative moiety of lignin molecule containing β–O–4 linkages (43–65% in lignin)[66], $4CeO_x/Ni$ catalyst also demonstrated high activity (83% yield) in the hydrogenolysis of VG to produce the corresponding 3–(3,4–dimethoxyphenyl)–1–propanol product by cleavage of the etheric β–O–4 bond and the neighboring internal hydroxyl C–O bond, which was much higher than that of 26% yield over conventional $96CeO_x/Ni$ catalyst. Moreover, this $4CeO_x/Ni$-catalyzed reductive scission of etheric C–O bond using $H_2$ was also found to be a powerful strategy to depolymerize polyether. Using polyethylene glycol (PEG–200) as the model polyether, the corresponding depolymerization product, ethanol, was obtained in 100% atom-efficiency with up to 85% yield under similar reaction conditions with the hydrogenolysis of bio-derived molecules. $4CeO_x/Ni$ also depolymerized the poly(bisphenol A-co-epichlorohydrin) glycidyl end-capped (PBAE) via selective hydrogenolysis of etheric C–O bond, achieving a 53% yield of propane–2,2–diyldicyclohexane and 4–(2–cyclohexylpropan–2–yl) phenol. In addition, the $4CeO_x/Ni$ catalyst demonstrated high catalytic activity in the hydrogenation conversion of polyethylene terephthalate (PET) to dimethyl cyclohexane dicarboxylate in 80% yield. These results show the great potential of such inverse $4CeO_x/Ni$ catalyst in the recycle of plastics waste.

## Discussion

In summary, the inverse catalyst of $4CeO_x/Ni$ demonstrated a remarkable enhancement in catalytic activity and selectivity for hydrogenolysis of various etheric C–O bonds in lignocellulose derivatives and polyether to produce valuable hydroxyl compounds with an atom-economic manner. Furthermore, the stability of this non-precious metal catalyst was well verified in the ring-opening hydrogenation of THFA to 1,5–PDO in a fixed-bed reactor with unprecedented activity under solvent-free condition. The systematic experimental and computational studies revealed that the obvious advantage of inverse $4CeO_x/Ni$ catalyst over the conventional $96CeO_x/Ni$ catalyst was enabled by the strong interfacial electronic interaction between the oxide and metal, which promoted the cleavage of etheric C–O bond through the enhanced adsorption of etheric oxygen adsorbed at the electron-rich interfacial Ce sites adjacent to oxygen vacancy, and dominated the formation of high content of $H^{\delta-}$ species adsorbed on interfacial Ni sites over the

inverse catalyst, both cooperatively boosted the reaction rate of C−O bond hydrogenolysis. This work advanced the fundamental understanding of interfacial electronic interaction over oxide/metal inverse catalyst and provided a promising catalyst design strategy for the efficient transformation of biomass and oxygen-containing polymer waste.

## Methods

### Chemicals

Nickel nitrate hexahydrate ($Ni(NO_3)_2 \cdot 6H_2O$, 99%), cobalt nitrate hexahydrate ($Co(NO_3)_2 \cdot 6H_2O$, 99%), copper nitrate trihydrate ($Cu(NO_3)_2 \cdot 3H_2O$, 99%), cerium nitrate hexahydrate ($Ce(NO_3)_3 \cdot 6H_2O$, 99.9%), ammonium perrhenate ($NH_4ReO_4$, 99.9%), ammonium metavanadate ($NH_4VO_3$, 99%), ammonium paratungstate ($(NH_4)_{10}H_2(W_2O_7)_6$, >99%), ammonium molybdate tetrahydrate ($(NH_4)_6Mo_7O_{24} \cdot 4H_2O$, >99%), oxalic acid (99%), tetrahydrofurfuryl alcohol (THFA, 99%), 5−hydroxymethyl furfural (97%), veratrylglycerol-β-guaiacyl ether (97%), polyethylene glycol (PEG, $M_V = 200$), PET (white granular, intrinsic viscosity of 0.82 dL/g), and PBAE (Mn ≈ 1075) were purchased from Adamas Reagent Ltd. All the chemicals and reagents were used as received without further purification, and the corresponding aqueous solutions were prepared with DI water (18.25 MΩ cm).

### Preparation of nCeOₓ/Ni catalysts

A series of Ni-supported $CeO_2$ catalysts with different $CeO_2$ loading (denoted as $nCeO_x/Ni$, $n = 1$, 4, 85, and 96) were prepared by the coprecipitation method. Typically, for the $4CeO_2/Ni$ catalyst, 200 mL of aqueous solution containing 23.2 g of $Ni(NO_3)_2 \cdot 6H_2O$ and 0.43 g of $Ce(NO_3)_3 \cdot 6H_2O$ was added dropwise into 200 mL of the oxalic acid solution (0.8 mol $L^{-1}$) under vigorous stirring. After being continuously stirred for one more hour, the mixture was sealed in a 500 mL Teflon bottle and hydrothermally treated in a stainless-steel autoclave at 140 °C for 24 h. The filter cake was washed with DI water three times. Then, the dried sample was calcined at 400 °C for 3 h in the air. The obtained powder was denoted as the calcined catalyst. Before the reaction, the calcined catalyst was reduced in a $H_2/N_2$ flow (10 vol %) at 350 °C for 3 h with a controlled heating rate of 2 °C $min^{-1}$ in the fixed-bed reactor. For the preparation of $96CeO_x/Ni$ catalyst, 1.53 g of $Ni(NO_3)_2 \cdot 6H_2O$ and 21.7 g of $Ce(NO_3)_3 \cdot 6H_2O$ were used as the precursors. Similarly, for the $1CeO_x/Ni$ catalyst, 23.2 g of $Ni(NO_3)_2 \cdot 6H_2O$ and 0.11 g of $Ce(NO_3)_3 \cdot 6H_2O$ were used. The $85CeO_x/Ni$ catalyst was prepared by precipitating 5.81 g of $Ni(NO_3)_2 \cdot 6H_2O$ and 17.4 g of $Ce(NO_3)_3 \cdot 6H_2O$ with oxalic acid.

### Preparation of physical mixed PM-4CeO₂ + Ni catalyst

For the preparation of the physical mixed catalyst of $PM-4CeO_2 + Ni$, $CeO_2$ and NiO powders were first prepared by the coprecipitation method, using $Ce(NO_3)_3 \cdot 6H_2O$ (23.2 g) and $Ni(NO_3)_2 \cdot 6H_2O$ (23.3 g) as precursors and oxalic acid as the precipitant. After drying and calcination, the resulting $CeO_2$ (0.29 g) and NiO (10.0 g) were physically mixed in a molar ratio of 1/80 (Ce to Ni) using an agate mortar. Finally, the mixture was reduced by $H_2/N_2$ in a fixed-bed reactor before the hydrogenation reaction.

### Preparation of IM-M'Oₓ/Ni catalysts

The catalysts of IM-M'Oₓ/Ni (M' = Ce, Re, Mo, V, and W) were prepared by the impregnation method (IM). The corresponding precursors, such as $Ce(NO_3)_3 \cdot 6H_2O$ (0.073 g), $NH_4ReO_4$ (0.045 g), $NH_4VO_3$ (0.019 g), $(NH_4)_{10}H_2(W_2O_7)_6$ (0.043 g), $(NH_4)_6Mo_7O_{24} \cdot 4H_2O$ (0.029 g), were respectively loaded on the prepared NiO (1.0 g) with the molar ratio of M' to Ni of 1/80. After calcined at 400 °C and subsequently reduced at 350 °C, the catalysts of IM-M'Oₓ/Ni were obtained.

### Structural characterization

High-resolution transmission electron microscopy images, high-angle annular dark field scanning transmission electron microscopy images,

and the energy dispersive X-ray spectroscopy mapping were obtained using an FEI F20 (200 kV). XRD experiments were performed on a Rigaku SmartLab diffractometer with Cu-Kα radiation. Quasi in situ XPS experiments were performed on a Thermo Scientific NEXSA Instrument. Before the XPS measurements, the catalyst was reduced in a glass tube, which was then sealed and transferred into a glove box without exposure to air. Subsequently, the sample was loaded on a sealed specimen stage and transferred into the XPS analysis chamber. All of the operations were carried out at room temperature inside a glove box. UV-Raman spectra were collected on a Labram HR Evolution (HORIBA) with a semiconductor laser (325 nm, 1 mW). The diffuse reflection ultraviolet-visible spectra were collected on a UV-vis spectrophotometer (Shimadzu, UV-2700). Quasi in situ EPR experiments were carried out on a JEOL RESONANCE JES-X320 spectrometer operating at X-band frequency ($v \approx 9.15$ GHz) at 25 °C. Before the EPR measurements, the catalyst powder was reduced in a glass tube, which was then sealed and transferred into a glove box without exposure to air. Subsequently, the sample was sealed in a glass capillary under a $N_2$ atmosphere. All of the operations were performed at room temperature inside a glove box. The metal content was determined by ICP-MS on a Thermo Fisher ICAP RQ instrument. The nitrogen adsorption-desorption isotherms were obtained on an ASAP 2020 Micromeritics instrument at 77 K, and the specific surface area was calculated using Brunauer-Emmett-Teller equation.

### In situ infrared Fourier transform measurements

The adsorption configuration of THFA on the catalysts was studied using in situ FT-IR experiments. The experiments were conducted on a Bruker infrared spectrometer (Vertex 70). Before measurements, the calcined catalyst was first reduced at 350 °C and then cooled to 50 °C in a 10 vol % $H_2/N_2$ flow. In the THFA hydrogenolysis process, the chamber was first evacuated to a vacuum. Then, saturated THFA gas was introduced into the chamber for 1 h. After the saturated adsorption of THFA on the catalyst surface, a 10 vol % $H_2/N_2$ flow (30 mL $min^{-1}$) was purged for 1 h to remove the physically adsorbed THFA molecule., and the corresponding IR spectra were collected. In the H−D exchange experiments, the catalyst was pre-reduced at 350 °C in a $H_2/N_2$ flow. After cooling to 50 °C under the Ar atmosphere, the chamber was purged by $D_2$ for 0.5 h. Subsequently, the sample was ramped up to 200 °C in a $D_2$ flow, and the corresponding IR spectra were monitored.

### Quasi in situ XPS spectra analysis for THFA adsorption analysis

Firstly, the reduced catalyst was transferred into a glove box without exposure to air. Then, the catalyst sample was impregnated in THFA for several minutes. After saturation adsorption of THFA, the catalyst was vacuumed to remove the physically adsorbed THFA. Subsequently, the catalyst was transferred into the XPS analysis chamber.

### H−D exchange experiments

First, the calcined catalyst was reduced at 350 °C in $H_2/N_2$ flow (30 mL $min^{-1}$, 10 vol % $H_2$) and then cooled to 50 °C. The chamber was purged by $D_2$ (15 mL $min^{-1}$) mixed with $H_2$ (15 mL $min^{-1}$) for 1 h. Subsequently, the sample was heated to 400 °C with a heating rate of 10 °C $min^{-1}$ in a mixture gas of $H_2 + D_2$, and the MS signal of HD (m/z = 3) was collected and analyzed with a mass spectrometer. The formation rate of HD was evaluated by the corresponding ion current intensity.

### Hydrogen temperature-programmed desorption

$H_2$-TPD experiments were performed on a TP5080 chemisorption analyzer (Tianjin Xianquan Co., Ltd, China) equipped with a mass spectrometry detector. Prior to $H_2$ desorption, the calcined catalysts were pretreated at 350 °C and then cooled down to 50 °C using a $H_2/N_2$ flow (30 mL $min^{-1}$, 10 vol % $H_2$). After the cooling process, the sample was purged for 1 h and then ramped up to 400 °C at a heating rate of

10 °C min$^{-1}$ in a He flow (30 mL min$^{-1}$). During the ramping process, the MS signals of H$_2$ (m/z = 2) were recorded.

## Carbon monoxide temperature-programmed desorption

CO-TPD was performed on a chemisorption analyzer equipped with a mass spectrometry detector. Before measurements, the catalysts were pre-reduced at 350 °C using a 10 vol % H$_2$/N$_2$, and then CO molecules were adsorbed at 30 °C for 0.5 h in a flow of 10 vol % CO/N$_2$. After purging the physically adsorbed CO molecules, the sample was heated at 10 °C min$^{-1}$ to 800 °C in Ar, and the MS signals of CO (m/z = 28) and CO$_2$ (m/z = 44) were recorded.

## Hydrogen temperature-programmed reduction

H$_2$-TPR was carried out with a chemisorption analyzer equipped with a thermal conductivity detector (TCD). Firstly, the catalysts were pre-treated at 400 °C for 1 h under He flow. After cooling down to 30 °C, the quartz tube reactor was purged with flowing H$_2$/N$_2$ (10 vol % H$_2$). The sample was ramped to 800 °C with a heating rate of 10 °C min$^{-1}$ in 10 vol % H$_2$/N$_2$ flow (30 mL min$^{-1}$), and the TCD signal was recorded.

## Hydrogenolysis of THFA in a fixed-bed reactor

The catalytic activity of the nCeO$_x$/Ni catalyst was evaluated in a fixed-bed reactor. Typically, 5.4 g of calcined 4CeO$_x$/Ni catalyst was formed into 40–60 mesh pellets. The pellets were then loaded into the stainless-steel tubular reactor with quartz liner (inner diameter of 10 mm). Before the reaction, the calcined catalyst was reduced at 350 °C using a 10 vol % H$_2$/N$_2$ flow (40 mL min$^{-1}$) for 3 h. After cooling down to the desired temperature, a feed of THFA (solvent-free, 0.02 mL min$^{-1}$) was pumped into the reactor, and the hydrogenolysis reaction was carried out at a temperature of 160 °C under flowing H$_2$ (4 MPa, 50 mL min$^{-1}$). During the reaction, the product was collected through a condenser pipe at −10 °C. The collected products were quantitatively analyzed using gas chromatography (GC) equipped with an FFAP chromatographic column and flame ionization detector (Shimadzu GC-2014C). The specific reaction rate was calculated using the following equation:

$$r = \frac{molar\ yield\ of\ 1,5-PDO\,(\mu mol)}{mass\ of\ catalyst(g) \times reaction\ time(\min)}$$

Here, the molar yield of 1,5-PDO was determined from the conversion of THFA and the selectivity of 1,5-PDO based on three independent measurements.

## Hydrogenation of the biomass compounds and polyether/ester in a batch reactor

For the reaction in a batch reactor, the catalyst was initially reduced at 350 °C for 3 h in a 10 vol % H$_2$/N$_2$ flow (50 mL min$^{-1}$) and subsequently passivated in a 5 vol% O$_2$/N$_2$ flow (40 mL min$^{-1}$) at 30 °C for 5 mins. Prior to the reaction, the autoclave was purged three times and then pressured with H$_2$ gas. The hydrogenolysis reaction of THFA (0.5 g) was catalyzed by a 0.25 g catalyst in a stainless-steel autoclave (100 mL). The reaction was conducted at a temperature of 160 °C under continuous stirring. The conversion of 5-hydroxymethyl furfural (0.5 g) was catalyzed by catalyst (0.25 g) under similar reaction conditions with the hydrogenolysis of THFA. For the transformation of veratrylglycerol-β-guaiacyl ether (0.25 g), the reaction (0.12 g of catalyst) was carried out at 150 °C. The depolymerization of polyethylene glycol (PEG–200, 0.5 g) was performed by using tetrahydrofuran as the solvent (0.4 g catalyst, 160 °C, 4.0 MPa H$_2$). The reaction of 0.5 g of PET (white granular, intrinsic viscosity of 0.82 dL/g) was carried out in methanol under 170 °C (0.4 g catalyst, 4.0 MPa H$_2$). The conversion of PBAE (Mn ≈ 1075) was carried out in methanol (0.4 g catalyst, 170 °C, 4.0 MPa H$_2$). The quantitative analysis of the obtained products was determined by GC.

## DFT calculation

First-principles calculations were performed using periodic DFT[67]. The spin-polarized generalized gradient approximation with the PBE functional was used to treat exchange-correlation effects. A plane wave basis set with a cutoff energy of 450 eV was selected to describe the valence electrons. The electron-ion interactions were described by the projector augmented wave method[68.] The Brillouin zone integration was performed with a 2 × 2 × 1 Monkhorst-Pack k-mesh. The SCF, the force convergence criteria for structural optimization and the convergence criteria of transition states were set to 1 × 10$^{-5}$ eV, 0.01 eV/Å and 0.03 eV/Å, respectively. The climbing image nudged elastic band and dimer methods were used to optimize the transition state structures. To describe the strongly correlated f-electrons of Ce, we adopt Dudarev's DFT + U scheme with $U$ = 5 eV, following previous research[69]. We used Grimme's DFT-D3 scheme to include the van der Waals interactions semi-empirically[70]. The optimized conventional cell of ceria with a lattice parameter of 3.871 Å was adopted to construct the oxygen-terminated CeO$_2$(111) surface. A (3 × 2√3) supercell with four O–Ce–O trilayers and 15 Å vacuum slab was used to describe the ceria support of the conventional catalyst during the catalytic process. An oxygen atom on the topmost layer of CeO$_2$(111) is removed to create oxygen vacancy. A Ni$_{10}$ cluster is adopted to represent the Ni particle, which occupies six surface oxygen sites after optimization. The inverse catalyst is modelled with a four-layer (6 × 6) Ni(111) slab supporting a Ce$_{10}$O$_{19}$ cluster. Considering the large size of the (6 × 6) Ni(111), only Γ point was used in the Brillouin zone sampling.

## Data availability

All data were available in the main text or the supplementasry materials. Source data of the figures are provided. Source data are provided with this paper.

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

## Acknowledgements

This work was funded by the National Natural Science Foundation of China (22102196, 22102193, 22472177, and 22272187), Jiangsu Province Science and Technology Project BK20211095, the Major Science and Technology Projects of Gansu Province (22ZD6GA003, 23ZDFA016), the Basic Research Creative Groups Project of the Science and Technology Plan of Gansu Province (23JRRA568).

## Author contributions

Z.Z. designed the study and wrote the manuscript. Z.Z. and G.G. performed most of the experiments and analyzed the results. Y.X. performed DFT calculations. J.W. conducted the FT-IR and isotope experiments. C.L. and Q.L. helped with the XPS and kinetic experiments. J.W., P.S., and Z.H. provided valuable discussions. F.L. designed and guided the study, and co-wrote the manuscript.

## Competing interests

The authors declare no competing interests.
