## [Peer Review File · Nature Communications]

Inverse ceria-nickel catalyst for enhanced C–O bond hydrogenolysis of biomass and polyetherREVIEWER COMMENTS

Reviewer #1 (Remarks to the Author):

The authors demonstrate that a novel inverse catalyst of CeOx nanoparticles supported on bulk Nickel was efficient for C-O bond hydrogenolysis on a variety of substrates. The authors synthesized a nice library of materials by coprecipitation to test the influence of relative ceria and nickel loadings. Extensive characterization by XPS, TEM, UV-Raman, EPR, XRD, confirmed the synthesis of both Ni/CeO₂ and CeOx/Ni catalysts and indicated that the different catalyst structures impacted the oxidation state of Ni and the oxygen vacancies on CeOx. In situ FTIR was then used to study the adsorption of THFA and H₂-D₂ exchange, concluding that the interfacial sites on the inverse catalyst were favorable for both processes. These analyses were supported by DFT and Bader charge analysis which are both outside of my area of expertise so I will not comment on these calculations. Hydrogen TPD analysis indicated that the inverse catalyst absorbed significantly more hydrogen than the other catalysts, and desorbed the hydrogen at lower temperatures. Finally, the authors demonstrated activity of the inverse vs conventional catalysts for the hydrogenolysis of a variety of model compounds representing lignin, cellulose, and polyether, showing that the inverse achieved significantly higher yields in all cases.

In my opinion, this report provides a unique catalyst for ether bond hydrogenolysis, demonstrates a promising activity and selectivity, and performs many rigorous analyses to characterize the structure. However, it is lacking some key controls and analyses to support the claims in the paper. With the additional analyses and controls suggested below, I think this work would be suitable for publication in nature communications.

A key conclusion of the paper is that the hydrogenation reactions are facilitated by the CeOx/Ni interfacial sites in the inverse catalyst and not on metallic Ni sites. Metallic Ni sites are also known to catalyze ether hydrogenolysis reactions. It is essential to provide relevant reactivity comparisons between the catalysts, normalizing by the different types of sites.

- The authors report a 36.5 fold increase in the reactive rate with the 4CeOx/Ni vs the 96CeOx/Ni in the abstract and introduction without noting how these rates are normalized. It is important to indicate whether this is based on total catalyst mass, total Nickel mass, or total number of active sites as measured by chemisorption. Based on the ICP-MS results in the SI, the 4CeOx/Ni material has 25.4x higher nickel loading than 96CeOx/Ni. Could this account for the increased rate?
- The rates in figure 1a are reported relative to the total mass of catalyst.
 - o Could the authors please also report the activity relative to the mass of nickel?
 - o Could the authors please also report the activity relative to the active surface area of the catalyst?
- ♣ The quantitative H₂ TPD analysis could provide an estimate of number of active sites. Alternatively CO chemisorption experiments would provide a measure of metallic sites.
- ♣ Is it possible to estimate the Ce/Ni interfacial area, for example based on the TEM analysis?
 - o I would recommend presenting each of these normalizations to provide the reader with a more complete picture of the relative activity of each catalyst.
- For metal/ceria catalysts, a useful characterization of the metal availability and metal-ceria interface is temperature programmed reduction. Ceria has characteristic surface and bulk reduction peaks, which can be shifted to lower temperatures when in contact with a metal. I think

that hydrogen TPR analysis would be a key characterization to support the claims about nickel/ceria interfacial differences between catalysts.

- It is important to perform catalyst stability studies at incomplete conversion in order to see any changes that occur to the activity over time. It is difficult to tell from the plot in figure 1b whether the reaction was at 100% conversion. Was there any of the unreacted THFA detected in the outlet of the reactor? Could the authors please report the values for initial and final THFA conversion during the 200h stability study?

Reviewer #2 (Remarks to the Author):

The presented work showcases an inverse ceria-nickel catalyst for hydrogenolysis of biomass-derived molecules and polyether, providing a promising attempt to explore the application of inverse catalyst in the precise transformation of complex molecules. The work successfully achieved a high conversion of THFA above 95% and 98% selectivity of 1,5-PDO over the inverse 4CeOx/Ni catalyst without using any solvent. The authors investigated the interfacial electronic interaction between CeOx and Ni via various characterizations and studied its relationship with catalytic performance. In general, the study was well-performed, and the manuscript is well-written, and the catalytic performance is impressive, which endows the manuscript potentially publishable in the journal of Nature Communications. However, before the manuscript can be considered for publication, there are still some small issues and concerns that need to be addressed.

1. The XPS results using Ar⁺ ion sputtering treatment require more detailed explanation to draw a clear conclusion. Additionally, the authors may want to explain the significant chemical valence change in the Ce species in the 96CeNi sample after Ar⁺ ion sputtering treatment, as shown in Figure S2d.
2. The details of measuring reaction rate were not provided, and it is unclear how the reaction rate was obtained. It would be helpful to know if they were measured at kinetic region by excluding the heat effect and mass transfer effect.
3. While the interfacial sites were identified as the active sites, a detailed understanding seems to be lacking. For example, the 4CeO₂/Ni sample with four times the CeO₂ amount has only 2 times the reaction rate of that of the 1CeO₂/Ni sample. This suggests that the 1CeO₂/Ni may have higher catalytic efficiency (TOF), provided that these two samples have similar CeOx size (thus the 4 times of interfacial site number). The authors may want to consider this aspect and have a discussion.
4. It is unclear what the actual active sites for the hydrogenolysis of ether are - metal site or the metal oxide site. Metallic metal sites are generally believed to be more effective for the hydrogenolysis, so why does it seem that the oxides are more effective in this work?
5. When comparing the production rate of 1,5-PDO, the content of Ni or Ce should also be considered. It is important to determine the critical active sites for the hydrogenolysis of ether - metal site or the metal oxide site.
6. In the EPR results, the shift of the g factor may not directly reflect the electron density on Ni.
7. The electron density of interfacial Ni species and Ce species in 4CeOx/Ni is lower than in

96CeOx/Ni. The definition and distinction of Ni at the interface should be elucidated.

8. While the writing is quite good, there are still a few poorly constructed sentences. For example, "While the strong diffraction peaks of 96CeOx/Ni and 85CeOx/Ni catalysts were assigned to the characteristic fluorite CeO₂ phase (PDF No: 65-5923)," needs to be revised.

Reviewer #3 (Remarks to the Author):

Li et al introduce via this study the use of inverse Ce/Ni based catalyst with controlling its initial composition. The authors claims that the 4CeOx/Ni catalyst is the most active for the upgrading of the oxygen containing biomass and plastic wastes and prob reactions for this study. The current research showcases the potential in adjusting the interfacial electronic structure of inverse catalysts help to show better activity than traditional metal-oxide supported catalysts for these reactions. However, it necessitates further elucidation and examination of additional aspects before consideration for publication in Nature Communications.

- Some observation was presented without an explanation in several key areas. For example, there was no clarification on why cerium (Ce) peaks were absent in the X-Ray Diffraction (XRD) for the 4CeOx/Ni, as illustrated in figures S1, with only nickel (Ni) characteristic peaks being observed. Additionally, the X-ray Photoelectron Spectroscopy (XPS) data require further elucidation concerning the observed trends, particularly why both Ni and Ce signals diminish in the 4CeOx/Ni catalyst, a trend not echoed in the 96CeOx/Ni catalyst.

- There's also a need to address the dimensions of the CeOx clusters within the 4CeOx/Ni and 1CeOx/Ni to compare these with the size of Ni in the conventional samples. It is important to evaluate how these observed sizes align with modeled structure by Density Functional Theory (DFT) modeling, potentially necessitating a discussion on size distribution and particle counts.

- Moreover, the reason behind the increase in oxygen vacancies in the 4CeOx/Ni following tetrahydrofuran (THF) adsorption. Similarly, an exploration is needed into why the 1CeOx/Ni sample exhibits a lower surface area and how this characteristic influences the catalytic activity.

- It is suggested that Energy Dispersive X-ray Spectroscopy (EDS) mapping be incorporated to monitor the dispersion of Ce and Ni atoms on the catalyst surfaces, which could provide valuable insights into the spatial distribution and interaction of these elements.

The authors assigned the enhanced of activity to the strong interfacial electronic interaction between the oxide and metal, which promoted the cleavage of etheric C–O bond.

- What about increase the ratio of the Ceria size would that increase the interface and thus the activity.

- Would other ratios like 10 or 20- CeOx/Ni still be a good candidate for the reaction as well?

Reviewer #4 (Remarks to the Author):

General:

In this study, the authors synthesized and characterized a catalyst comprised of ceria supported on nickel metal and employed it for carbon-oxygen bond hydrogenolysis of tetrahydrofurfural alcohol (THFA) and other model compounds. The “inverse catalyst”, denoted 4CeOx/Ni, was compared to the traditional ceria-supported nickel analogue, denoted 96CeOx/Ni, and found to have improved activity and selectivity. The authors performed a variety of complementary characterization techniques to support the theory that the 4CeOx/Ni has sufficient electronic structure to activate C-O bonds via oxygen vacancies and hydrogen on metallic sites, including FTIR, XPS, UV-Raman, UV-Vis, EPR, and H₂-TPD-MS. The catalysts were also characterized with TEM and ICP-MS. The “quasi in-situ” studies with XPS and EPR involved reduction of the catalysts followed by air-free transfer to the instrument, where the catalyst was saturated with THFA then vacuumed to only leave behind physisorbed material. Calculations based upon Density Functional Theory (DFT) were also used to compare the energetics of C-O bond cleavage over the different structures, suggesting the 4CeOx/Ni provides favorable adsorption at electron-rich Ce and Ni interfacial sites. The authors performed stability tests and used the catalyst for several other biomass-relevant model compounds including 5-hydroxymethyl furfural (HMF), veratrylglycerol-β-guaiacyl ether (VG), and polyethylene glycol (PEG). The implementation of the inverse supported catalyst for carbon-oxygen bond hydrogenolysis is timely and interesting from the perspective of bond activation. But overall, the application for biomass and plastic is very preliminary. Furthermore, several concerns regarding the activity comparison, stability experiments, quasi in-situ characterization, and model compounds should be addressed. If addressed, this paper might be suitable for publication in Nature Communications.

Specific:

1. For the literature presented in Supplementary Table 1, are the authors attributing the deactivation of Nickel-based catalysts to leaching, or general deactivation as evidenced by conversion loss? Is leaching a concern for the ceria-nickel based catalyst in this study?
2. The rates in Figure 1 are on a mass of catalyst basis- Did the authors compare selectivity and activity on a nickel basis? The mass basis might over-estimate activity.
3. The methods for the quasi in-situ XPS and EPR measurements are unclear. Are these measurements done at room temperature? What are the key differences between the conditions of the measurement and the actual experiment? What are the limitations of this measurement?
4. For stability measurements, the authors ran the 4CeOx/Ni catalyst for 200 hours for tetrahydrofurfuryl alcohol (THFA) conversion into 1,5-pentanediol (1,5-PDO). The authors used 13 gram of catalyst, and ran at 100% conversion. Given the large catalyst loading, and complete conversion, it is difficult to tell if the catalyst is truly stable or if the catalyst loading is so high that no loss in conversion is observed. It is recommended that the authors use a lower loading and operate at a lower conversion and lower spacetimes to characterize the stability of the catalyst.
5. For the model compound studies (HMF, VG, PEG), did the authors reduce the catalyst first and perform air-free transfer into the batch reactor vessel?
6. The authors compared the performance of the 4CeOx/Ni catalyst to the 96CeOx/Ni for the HMF, VG, and PEG substrates, with significantly lower yields over the 96CeOx/Ni. Given the 24-fold increase in nickel, in the reverse catalyst, how would the yields compare if the catalysts were normalized by Ni loading?

7. For PEG experiments, was there evidence that the solvent (THF) was converted?
8. Does PEG depolymerization occur over this catalyst in the absence of hydrogen? Would this catalyst be effective and/or practical for glycolysis/methanolysis/hydrolysis? Technically ethanol is not the monomer of PEG, it is ethylene glycol. What is the proposed mechanism here for ethanol formation?
9. Have the authors tried using polyethylene terephthalate (PET) or other polyesters for depolymerization? To make the claim of the utility of this catalyst beyond model compounds, the authors would have to demonstrate its applicability with additional polymeric materials.

Responses to Reviewers' comments

Reviewer #1

Comment. *The authors demonstrate that a novel inverse catalyst of CeO_x nanoparticles supported on bulk Nickel was efficient for C-O bond hydrogenolysis on a variety of substrates. The authors synthesized a nice library of materials by coprecipitation to test the influence of relative ceria and nickel loadings. Extensive characterization by XPS, TEM, UV-Raman, EPR, XRD, confirmed the synthesis of both Ni/CeO₂ and CeO_x/Ni catalysts and indicated that the different catalyst structures impacted the oxidation state of Ni and the oxygen vacancies on CeO_x. In situ FTIR was then used to study the adsorption of THFA and H₂-D₂ exchange, concluding that the interfacial sites on the inverse catalyst were favorable for both processes. These analyses were supported by DFT and bader charge analysis which are both outside of my area of expertise so I will not comment on these calculations. Hydrogen TPD analysis indicated that the inverse catalyst absorbed significantly more hydrogen than the other catalysts, and desorbed the hydrogen at lower temperatures. Finally, the authors demonstrated activity of the inverse vs conventional catalysts for the hydrogenolysis of a variety of model compounds representing lignin, cellulose, and polyether, showing that the inverse achieved significantly higher yields in all cases.*

In my opinion, this report provides a unique catalyst for ether bond hydrogenolysis, demonstrates a promising activity and selectivity, and performs many rigorous analyses to characterize the structure. However, it is lacking some key controls and analyses to support the claims in the paper. With the additional analyses and controls suggested below, I think this work would be suitable for publication in nature communications.

Response. We appreciate reviewer very much for the positive comment and encouragement.

Comment 1. *A key conclusion of the paper is that the hydrogenation reactions are facilitated by the CeO_x/Ni interfacial sites in the inverse catalyst and not on metallic Ni sites. Metallic Ni sites are also known to catalyze ether hydrogenolysis reactions. It is essential to provide relevant reactivity comparisons between the catalysts, normalizing by the different types of sites.*

Response. We thank the reviewer very much for this important reminder. As you suggested, we prepared typical Ni catalysts, including Ni/SiO₂^{R1}, Ni/Al₂O₃^{R2}, and Ni/C^{R3}, where metallic Ni was reported as the active sites for the hydrogenolysis of aryl ethers in the lignin model compounds. However, these Ni catalysts exhibited poor selectivity for 1,5-pentanediol (1,5-PDO) in the selective ring-opening hydrogenolysis of tetrahydrofurfuryl alcohol (THFA). Instead, methyltetrahydrofuran (MTHF) was obtained with selectivity over 50% (Table R1). While, the

4CeO_x/Ni catalyst with inverse structure afforded desired 1,5-PDO with up to 98% selectivity. These results demonstrated the metallic Ni sites was not the active sites for the hydrogenolysis of etheric C–O bond of THFA over inverse 4CeO_x/Ni catalyst.

Table R1. Selective hydrogenolysis of THFA over various catalysts in a batch reactor.^[a]

Catalyst	Conversion (%)	Selectivity (%)				Ref.
		1,5-PDO	1-butanol	1-pentanol	MTHF	
4CeO _x /Ni	100	98	<1	<1	<1	This work
Ni/SiO ₂	6	<1	<1	<1	66	R1
Ni/Al ₂ O ₃	15	<1	<1	<1	48	R2
Ni/C	10	<1	<1	<1	67	R3

[a] Reaction conditions: catalyst, 0.25 g; THFA, 0.5 g; temperature, 160 °C; ethanol, 5 mL; 4.0 MPa H₂; 24 h. MTHF = methyltetrahydrofuran

- R1. He, J., Zhao C. & Lercher J. A. Ni-catalyzed cleavage of aryl ethers in the aqueous phase. *J. Am. Chem. Soc.* **134**, 20768-20775 (2012).
- R2. Qi, L., *et al.* Unraveling the dynamic network in the reactions of an alkyl aryl ether catalyzed by Ni/γ-Al₂O₃ in 2-propanol. *J. Am. Chem. Soc.* **141**, 17370-17381 (2019).
- R3. Song, Q., *et al.* Lignin depolymerization (LDP) in alcohol over nickel-based catalysts via a fragmentation–hydrogenolysis process. *Energy Environ. Sci.* **6**, 994-1007 (2013).

To provide a more complete picture of the relative activity of each catalyst, the reaction rate of THFA hydrogenolysis over nCeO_x/Ni catalysts normalized by their interfacial perimeter length, surface area, Ce mass, and Ni mass were also provided correspondingly in **Supplementary Fig. 4** (for corresponding details about every normalization, please see the following responses to your comments). These results all indicated that the representative inverse 4CeO_x/Ni catalyst exhibited higher catalytic activity compared to the representative conventional 96CeO_x/Ni catalyst.

Supplementary Figure 4. The reaction rate of THFA hydrogenolysis over $n\text{CeO}_x/\text{Ni}$ catalyst normalized by their (a) interfacial perimeter length, (b) surface areas, (c) Ce mass, and (d) Ni mass.

The above results and the corresponding discussions were added in the **Main Text** as follows:

“To provide more information about the relationship between C–O bond hydrogenolysis activity and interface Ce–Ni sites, the reaction rate was normalized by their interfacial perimeter length, surface area, Ce mass, and Ni mass (Supplementary Fig. 4), which generally demonstrated the representative inverse $4\text{CeO}_x/\text{Ni}$ catalyst exhibited superior catalytic activity compared to the representative conventional $96\text{CeO}_x/\text{Ni}$ catalyst.³⁵”

Comment 2. *The authors report a 36.5 fold increase in the reactive rate with the $4\text{CeO}_x/\text{Ni}$ vs the $96\text{CeO}_x/\text{Ni}$ in the abstract and introduction without noting how these rates are normalized. It is important to indicate whether this is based on total catalyst mass, total Nickel mass, or total number of active sites as measured by chemisorption. Based on the ICP-MS results in the SI, the $4\text{CeO}_x/\text{Ni}$ material has 25.4x higher nickel loading than $96\text{CeO}_x/\text{Ni}$. Could this account for the*

increased rate?

Response. Thank you for this kind reminder. The reported reaction rate in the abstract and introduction was normalized by the total catalyst mass, and the corresponding reaction rate was calculated using the following equation:

$$r = \frac{\text{molar yield of 1,5-PDO } (\mu\text{mol})}{\text{mass of catalyst (g)} \times \text{reaction time (min)}}$$

As suggested, the experiment detail was added in the **Method** of **Main Text** as follows:

“The collected products were quantitatively analyzed using gas chromatography (GC) equipped with an FFAP chromatographic column and flame ionization detector (Shimadzu GC-2014C). The specific reaction rate was calculated using the following equation:

$$r = \frac{\text{molar yield of 1,5 - PDO } (\mu\text{mol})}{\text{mass of catalyst (g)} \times \text{reaction time (min)}}$$

Here, the molar yield of 1,5-PDO was determined from the conversion of THFA and the selectivity of 1,5-PDO based on three independent measurements.”

As shown in **Fig. 1a in main text**, trace or no 1,5-PDO was obtained in the THFA hydrogenolysis over bulk Ni or CeO₂ alone, indicating that the catalytic sites located at the Ce-Ni interface rather than on the CeO₂ or Ni. Combining with the results in **Table R1**, it could be clarified that the metallic Ni sites was not the active sites for the hydrogenolysis of etheric C–O bond in THFA over inverse 4CeO_x/Ni catalyst. Therefore, it was deduced that a 36.5-fold increase in the reactive rate with the 4CeO_x/Ni vs the 96CeO_x/Ni could not be attributed to the 25.4x higher nickel loading.

Comment 3. *The rates in figure 1a are reported relative to the total mass of catalyst. Could the authors please also report the activity relative to the mass of nickel?*

Response. Trace or no 1,5-PDO was obtained in the THFA hydrogenolysis over bulk Ni or CeO₂ alone, indicating that the actual catalytic sites located at the Ce-Ni interface rather than on the CeO₂ or Ni. In addition, the results in **Table R1** indicated that the metallic Ni sites was not the active sites for the hydrogenolysis of etheric C–O bond. Hence, the reaction rate normalized by the Ni mass might not be appropriate for understanding the difference in catalytic performance between the inverse and conventional interfacial Ce-Ni sites. Even so, as suggested, the reaction rate of nCeO_x/Ni for the THFA hydrogenolysis normalized by Ni mass was added in the **Supplementary Fig. 4d of Supporting Information**. The representative inverse 4CeO_x/Ni catalysts also exhibited a higher 1,5-PDO production rate than the representative conventional 96CeO_x/Ni catalysts (as

shown in **Supplementary Fig. 4d**).

Supplementary Figure 4. The reaction rate of THFA hydrogenolysis over $n\text{CeO}_x/\text{Ni}$ catalyst normalized by their (a) interfacial perimeter length, (b) surface areas, (c) Ce mass, and (d) **Ni mass**.

Comment 4. *Could the authors please also report the activity relative to the active surface area of the catalyst?*

Response. As suggested, the reaction rate of $n\text{CeO}_x/\text{Ni}$ for the THFA hydrogenolysis normalized by surface area determined by BET measurements (**Supplementary Table 2**) was added in the **Supplementary Fig. 4b** of **Supporting Information**. Generally, the catalyst possessed larger surface area afforded higher catalytic reaction rate. However, the $1\text{CeO}_x/\text{Ni}$ and $4\text{CeO}_x/\text{Ni}$ catalysts exhibited a higher 1,5-PDO production rate than the $85\text{CeO}_x/\text{Ni}$ and $96\text{CeO}_x/\text{Ni}$ catalysts, even though the specific surface area of $1\text{CeO}_x/\text{Ni}$ ($2 \text{ m}^2 \text{ g}^{-1}$) and $4\text{CeO}_x/\text{Ni}$ ($15 \text{ m}^2 \text{ g}^{-1}$) were smaller than those of $85\text{CeO}_x/\text{Ni}$ ($71 \text{ m}^2 \text{ g}^{-1}$) and $96\text{CeO}_x/\text{Ni}$ ($92 \text{ m}^2 \text{ g}^{-1}$). These results indicated the superior catalytic activity of the inverse interfacial Ce-Ni sites compared to its conventional counterpart.

Supplementary Figure 4. The reaction rate of THFA hydrogenolysis over nCeO_x/Ni catalyst normalized by their (a) interfacial perimeter length, (b) surface areas, (c) Ce mass, and (d) Ni mass.

Comment 5. *The quantitative H₂ TPD analysis could provide an estimate of number of active sites. Alternatively, CO chemisorption experiments would provide a measure of metallic sites.*

Response. Thank you for this helpful suggestion. As suggested, the carbon monoxide temperature-programmed desorption combined with mass spectrometry (CO-TPD-MS) of nCeO_x/Ni were conducted, and the signal of CO ($m/z_{\text{CO}}=28$) and CO₂ ($m/z_{\text{CO}_2}=44$) were recorded. The corresponding CO-TPD-MS profiles were added in **Supplementary Fig. 5** of **Supporting Information**. As shown in **Supplementary Fig. 5a**, the strong peaks at 250 °C and 600 °C could be attributed to the desorption of CO on the metallic Ni sites of conventional 96CeO_x/Ni catalyst (Gould, T. D., et al. *J. Catal.* **303**, 9-15 (2013)). However, no remarkable CO signal was observed over the 4CeO_x/Ni and 1CeO_x/Ni catalysts, which indicated the difference in the characteristics of metallic sites on the inverse catalyst, rather than the difference in the number of sites, compared to those on the conventional catalyst, because the Ni content in 4CeO_x/Ni and 1CeO_x/Ni is higher than that in 96CeO_x/Ni. Due to the negligible CO signal, it is challenging to estimate the number of metallic sites on 4CeO_x/Ni and 1CeO_x/Ni catalysts using CO-TPD-MS. During the process of CO desorption, CO molecule could be transformed into CO₂ through the oxidation by mobile lattice oxygen of CeO₂ (Saw, E. T., et al. *J. Catal.* **314**, 32-46 (2014)). As shown in **Supplementary Fig. 5b**, the oxidation of CO into CO₂ by CeO_x species occurred at a lower temperature of 200 °C over the inverse 4CeO_x/Ni catalysts, as compared to the temperature of 550 °C required over the conventional 96CeO_x/Ni catalyst, which could be ascribed to the weakened Ce–O bond in the inverse catalyst.

The corresponding discussion was added in the **Main Text** as follows:

“The carbon monoxide temperature-programmed desorption combined with mass spectrometry (CO-TPD-MS) results (Supplementary Fig. 5) of $n\text{CeO}_x/\text{Ni}$ catalysts indicated a remarkable difference in the surface metallic sites between inverse and conventional catalyst because of the negligible CO desorption for the $4\text{CeO}_x/\text{Ni}$ catalysts, compared to the strong desorption peaks at $250\text{ }^\circ\text{C}$ and $600\text{ }^\circ\text{C}$ for the $96\text{CeO}_x/\text{Ni}$ catalyst.³⁶ Moreover, the oxidation of CO into CO_2 by CeO_x species occurred at a lower temperature of $200\text{ }^\circ\text{C}$ over the $4\text{CeO}_x/\text{Ni}$ catalysts, as compared to the temperature of $550\text{ }^\circ\text{C}$ required over the $96\text{CeO}_x/\text{Ni}$ catalyst, which could be ascribed to the weakened Ce-O bond in inverse catalyst.³⁷ Based on the above results, it could be preliminarily deduced that the difference in the geometrical and electronic structure of inverse and conventional interfacial Ce-Ni sites contributed to the superior catalytic activity of the inverse catalyst compared to its conventional counterpart in the THFA hydrogenolysis.

36. Gould, T. D., et al. Synthesis of supported Ni catalysts by atomic layer deposition. *J. Catal.* **303**, 9-15 (2013).

37. Saw, E. T., et al. Bimetallic Ni-Cu catalyst supported on CeO_2 for high-temperature water-gas shift reaction: Methane suppression via enhanced CO adsorption. *J. Catal.* **314**, 32-46 (2014).”

The corresponding CO-TPD-MS profiles were added in **Supplementary Fig. 5** of **Supporting Information** as follows:

Supplementary Figure 5. CO-TPD-MS profiles of $n\text{CeO}_x/\text{Ni}$ catalysts. (a) CO signal of $m/z=28$; (b) CO_2 signal of $m/z=44$.

Comment 6. *Is it possible to estimate the Ce/Ni interfacial area, for example based on the TEM analysis?*

Response. We thank the reviewer very much for this helpful suggestion. Trace or no desired 1,5-PDO was obtained in the THFA hydrogenolysis over bulk Ni or CeO₂ alone, indicating that the actual catalytic sites located at the Ce-Ni interface. As reported in the literatures (Chen, A., et al. *Nat. Catal.* **2**, 334-341 (2019), Yan, H., et al. *Nat. Commun.* **10**, 3470 (2019)), the reaction rate was found to be a function of the interfacial perimeter length. Therefore, it could be deduced that the reaction rate of interfacial Ce-Ni sites in the THFA hydrogenolysis might also be associated with the interfacial perimeter length, which could be estimated from the particle size of the CeO_x or Ni clusters on the inverse and conventional CeO_x/Ni catalysts as determined by HRTEM analysis. The calculation detail was added in the **Supporting Information** as follows:

Supplementary Table 2. Physical and chemical characterizations of nCeO_x/Ni catalysts.

Sample	Ni content ^a (wt %)	Ce content ^a (wt %)	S _{BET} ^b (m ² g ⁻¹)	V _{Pore} ^b (m ³ g ⁻¹)	D _{Pore} ^b (nm)	Interfacial perimeter length ^c (m g ⁻¹)
96CeO _x /Ni	3.8	78.3	92	0.11	4	1.5×10 ⁹
85CeO _x /Ni	14.6	69.5	71	0.09	5	3.1×10 ⁹
4CeO _x /Ni	96.5	2.9	15	0.11	30	7.5×10 ⁹
1CeO _x /Ni	99.1	0.8	2	0.03	73	4.2×10 ⁹

^aActual Ni and Ce contents were determined by ICP-MS. ^bDetermined by N₂ physisorption. ^cThe total length of the nickel-ceria interfacial perimeter was calculated from the size of CeO_x clusters on inverse 1CeO_x/Ni and 4CeO_x/Ni catalysts, and Ni clusters on conventional catalyst, respectively.¹⁵

The perimeter length was calculated using the following equation:

$$Perimeter\ length = \frac{m_{CeO_x/Ni} \times \pi \times d_{CeO_x/Ni}}{\rho_{CeO_x/Ni} \times \frac{1}{2} \times \frac{4}{3} \times \pi \times \left(\frac{d_{CeO_x/Ni}}{2}\right)^3}$$

Here, the model of hemispherical CeO_x or Ni cluster supported on Ni or CeO₂ was used. $d_{CeO_x/Ni}$ represented the particle size of CeO_x or Ni cluster determined from the HRTEM analysis. $m_{CeO_x/Ni}$ and $\rho_{CeO_x/Ni}$ was the loading content of and the density of CeO₂ or Ni, respectively.

15. Yan, H., et al. Construction of stabilized bulk-nano interfaces for highly promoted inverse CeO₂/Cu catalyst. *Nat. Commun.* **10**, 3470 (2019).”

Comment 7. I would recommend presenting each of these normalizations to provide the reader with a more complete picture of the relative activity of each catalyst.

Response. Thank you for this suggestion. As a summary of above several responses and following the suggestion in this comment, the complete normalization of the reaction rate of THFA hydrogenolysis over $n\text{CeO}_x/\text{Ni}$ catalysts with their corresponding interfacial perimeter length, surface area, Ce mass, and Ni mass was provided as illustrated in **Supplementary Fig. 4**. These results generally indicated that the representative inverse $4\text{CeO}_x/\text{Ni}$ catalyst display higher catalytic activity in the hydrogenolysis of THFA compared to the representative conventional $96\text{CeO}_x/\text{Ni}$ catalyst. The results were added in the **Supplementary Fig. 4** of **Supporting Information** as follows:

Supplementary Figure 4. The reaction rate of THFA hydrogenolysis over $n\text{CeO}_x/\text{Ni}$ catalyst normalized by their (a) interfacial perimeter length, (b) surface areas, (c) Ce mass, and (d) Ni mass.

The corresponding discussion was added in the **Main Text** as follows:

“To provide more information about the relationship between C–O bond hydrogenolysis activity and interface Ce-Ni sites, the reaction rate was normalized by their interfacial perimeter length, surface area, Ce mass, and Ni mass (Supplementary Fig. 4), which generally demonstrated the representative inverse 4CeO_x/Ni catalyst exhibited superior catalytic activity compared to the representative conventional 96CeO_x/Ni catalyst.^{33,35}

33. Yan, H., et al. Construction of stabilized bulk-nano interfaces for highly promoted inverse CeO₂/Cu catalyst. *Nat. Commun.* **10**, 3470 (2019).

35. Chen, A., et al. Structure of the catalytically active copper-ceria interfacial perimeter. *Nat. Catal.* **2**, 334-341 (2019).”

Comment 8. For metal/ceria catalysts, a useful characterization of the metal availability and metal-ceria interface is temperature programmed reduction. Ceria has characteristic surface and bulk reduction peaks, which can be shifted to lower temperatures when in contact with a metal. I think that hydrogen TPR analysis would be a key characterization to support the claims about nickel/ceria interfacial differences between catalysts.

Response. Thank you for the helpful suggestions. As suggested, the H₂-TPR experiments were carried out. As shown in **Supplementary Fig. 8c**, the peaks at 130 °C for the inverse 4CeO_x/Ni and 1CeO_x/Ni catalysts were attributed to the reduction of surface adsorbed oxygen species associated with the formation of interfacial Ni–V_O–Ce (Liao, X., et al., *Appl. Catal., A* **488**, 256-264 (2014)). In contrast, no remarkable reduction peak between 40 °C and 200 °C was observed over the conventional 85CeO_x/Ni and 96CeO_x/Ni catalysts. According to the reported literatures, the reduction temperature of interfacial Ce species for the conventional CeO_x/Ni catalysts could be within the range from 200 °C to 350 °C (Zhao, Z., et al. *Chem* **8**, 1034-1049 (2022)). These peaks could be overlapped with the broad reduction peaks of NiO to metallic Ni, observed at around 320 °C (Ang, M. L., et al. *ACS Catal.* **4**, 3237-3248 (2014)). Therefore, the lower reduction temperature of interfacial Ce species on the 4CeO_x/Ni and 1CeO_x/Ni catalysts could be attributed to the stronger interaction between interfacial Ni and Ce species on the inverse catalyst (Zhang, S., et al. *Nanoscale* **9**, 3140-3149 (2017)).

As suggested, the results of the H₂-TPR were added in the **Supplementary Fig. 8** of **Supporting Information** as follows:

Supplementary Figure 8. Quasi in situ XPS spectra of (a) Ce 3d and (b) Ni 2p spectra of nCeO_x/Ni catalysts. (c) H₂-TPR profiles and (d) EPR spectra of the nCeO_x/Ni catalysts.

The corresponding experimental detail and discussion were added in the **Main text** as follows:

“H₂-TPR was carried out with a chemisorption analyzer equipped with a thermal conductivity detector (TCD). Firstly, the catalysts were pretreated at 400 °C for 1 h under He flow. After cooling down to 30 °C, the quartz tube reactor was purged with flowing H₂/N₂ (10 vol % H₂). The sample was ramped to 800 °C with a heating rate of 10 °C min⁻¹ in 10 vol % H₂/N₂ flow (30 mL min⁻¹), and the TCD signal was recorded.”

“...Hydrogen temperature-programmed reduction (H₂-TPR) experiments were performed to study the interaction between Ni and Ce species on the nCeO_x/Ni catalysts. As shown in Supplementary Fig. 8c, the peaks at 130 °C for the inverse 4CeO_x/Ni and 1CeO_x/Ni catalysts were attributed to the reduction of surface adsorbed oxygen species associated with the formation of interfacial Ni–V_O–Ce.⁴⁵ In contrast, no remarkable reduction peak between 40 °C and 200 °C was observed over the conventional 85CeO_x/Ni and 96CeO_x/Ni catalysts, indicating that the reduction of interfacial Ni–O–Ce species over the conventional catalyst could occur at higher temperature, and the corresponding reduction peak might be overlapped with the broad peak at around 320 °C belonged to the reduction of NiO to metallic Ni.⁴⁶ The low reduction temperature of Ni–V_O–Ce species on the 4CeO_x/Ni and 1CeO_x/Ni catalysts indicated the strong interaction between interfacial Ni and Ce species on inverse catalyst,⁴⁷...”

45. Liao, X., Zhang Y., Hill M., Xia X., Zhao Y. & Jiang Z. Highly efficient Ni/CeO₂ catalyst for the liquid phase hydrogenation of maleic anhydride. *Appl. Catal., A* **488**, 256-264

(2014).

46. Ang, M. L., *et al.* Highly active Ni/xNa/CeO₂ catalyst for the water–gas shift reaction: Effect of sodium on methane suppression. *ACS Catal.* **4**, 3237-3248 (2014).
47. Zhang, S., *et al.* Towards highly active Pd/CeO₂ for alkene hydrogenation by tuning Pd dispersion and surface properties of the catalysts. *Nanoscale* **9**, 3140-3149 (2017).”

Comment 9. *It is important to perform catalyst stability studies at incomplete conversion in order to see any changes that occur to the activity over time. It is difficult to tell from the plot in figure 1b whether the reaction was at 100% conversion. Was there any of the unreacted THFA detected in the outlet of the reactor? Could the authors please report the values for initial and final THFA conversion during the 200h stability study?*

Response. Thank you for this helpful suggestion. As shown in **Fig. 1b in main text**, the inverse 4CeO_x/Ni catalyst exhibited high stability in the hydrogenolysis of THFA, affording 95% conversion of THFA during the initial 20 h and maintaining 95% at the final of 200 h, respectively. To better study the catalyst’s stability, the controlled reaction with about half conversion of THFA was conducted under the similar reaction condition, except for the increase in the feed rate of THFA (solvent-free) from 0.02 mL min⁻¹ to 0.04 mL min⁻¹ (calcinated catalyst: 13 g; 160 °C; 4.0 MPa H₂; gas flow rate: 50 mL min⁻¹). As shown in **Supplementary Fig. 6**, the catalyst also worked stably for over 200 h, affording >97% selectivity of 1,5-PDO under the THFA conversion of about 55%. These results indicated the catalyst was stable for the hydrogenolysis of THFA under the reaction conditions. As suggested, the result was added to the **Supporting Information** as **Supplementary Figure 6** as follows:

Supplementary Figure 6. Catalyst stability experiment in a fixed-bed reactor over a 4CeO_x/Ni catalyst. Before the reaction, the calcinated catalyst was pre-reduced in a H₂/N₂ flow. Reaction conditions: calcinated catalyst: 13 g; THFA (solvent-free): 0.04 mL min⁻¹; 160 °C; 4.0 MPa H₂; gas-flow rate: 50 mL min⁻¹.

The corresponding description and discussion were added in the **Main Text** as follows:

“Notably, the inverse $4\text{CeO}_x/\text{Ni}$ catalyst also demonstrated good stability for 200 h in a fixed-bed reactor without remarkable decrease in the THFA conversion or 1,5-PDO selectivity at half and almost full conversion of THFA (Fig. 1b and Supplementary Fig. 6), which could be attributed to the inhibited agglomeration of metallic Ni species (Supplementary Fig. 7a)³⁸ and the enhanced stability in the Ce^{3+} species of CeO_x cluster (Supplementary Fig. 7b)⁵.”

Reviewer #2

Comment. *The presented work showcases an inverse ceria-nickel catalyst for hydrogenolysis of biomass-derived molecules and polyether, providing a promising attempt to explore the application of inverse catalyst in the precise transformation of complex molecules. The work successfully achieved a high conversion of THFA above 95% and 98% selectivity of 1,5-PDO over the inverse 4CeO_x/Ni catalyst without using any solvent. The authors investigated the interfacial electronic interaction between CeO_x and Ni via various characterizations and studied its relationship with catalytic performance. In general, the study was well-performed, and the manuscript is well-written, and the catalytic performance is impressive, which endows the manuscript potentially publishable in the journal of Nature Communications. However, before the manuscript can be considered for publication, there are still some small issues and concerns that need to be addressed.*

Response. We appreciate the reviewer very much for your positive comment and encouragement.

Comment 1. *The XPS results using Ar⁺ ion sputtering treatment require more detailed explanation to draw a clear conclusion. Additionally, the authors may want to explain the significant chemical valence change in the Ce species in the 96CeNi sample after Ar⁺ ion sputtering treatment, as shown in Figure S2d.*

Response. Thank you for this helpful suggestion. Ar⁺ ion sputtering is a powerful surface analysis method to obtain a depth profile of elements in a catalyst. In this work, we tried to compare the XPS signals before and after Ar⁺ ion sputtering to study the distribution of Ni and Ce elements from surface to the bulk in the inverse 4CeO_x/Ni and conventional 96CeO_x/Ni catalysts, respectively, which would help us to clarify their structure features. As shown in **Supplementary Figure 3a and b**, for the inverse 4CeO_x/Ni catalyst, the intensity of Ni peak at 852.3 eV (refer to baseline) increased from 2.0×10^5 to 4.2×10^5 counts after sputtering for 100 seconds. Simultaneously, the corresponding Ce element signal decreased from 1.6×10^4 to 0.7×10^4 counts (calculated based on the characteristic peak at 885.1 eV). The trend indicated that the relative amount of Ni elements increased from surface to sub-surface of the inverse 4CeO_x/Ni catalyst after Ar⁺ ion sputtering treatment, while the amount of Ce elements correspondingly decreased (**Scheme R1**). Similarly, according to the results in **Supplementary Figure 3c and d**, the Ni signal (peak at 852.3 eV) decreased from 0.7×10^5 to 0.3×10^5 counts over the 96CeO_x/Ni catalysts, and the corresponding Ce signal (peak at 882.3 eV) increased from 0.8×10^5 to 1.2×10^5 counts, which showed that the relative amount of Ce elements increased from surface to sub-surface of the conventional 96CeO_x/Ni catalyst after Ar⁺ ion sputtering treatment, while the amount of Ni elements correspondingly decreased (**Scheme R1**). The above results disclosed that Ce elements were scattered on bulk Ni surface of inverse 4CeO_x/Ni catalyst, while Ni elements were distributed on the CeO₂ surface of conventional 96CeO_x/Ni catalyst.

Scheme R1. Structural changes of the inverse $4\text{CeO}_x/\text{Ni}$ and conventional $96\text{CeO}_x/\text{Ni}$ catalysts after Ar^+ ion sputtering treatment.

In order to improve the readability, as suggested, we have revised the corresponding description in the **Main Text** as follows:

“Their geometrical structures were further studied by using element analysis through in situ X-ray photoelectron spectroscopy (XPS) combined with Ar^+ ion sputtering treatment (Supplementary Fig. 3). The intensity of Ni peak at 852.3 eV for inverse $4\text{CeO}_x/\text{Ni}$ catalyst increased from 2.0×10^5 to 4.2×10^5 counts after 100 seconds of sputtering. Simultaneously, the intensity of the Ce peak at 885.1 eV diminished from 1.6×10^4 to 0.7×10^4 counts. In contrast, after sputtering, the enhancement of Ce signals and the weakness of Ni signals were observed over conventional $96\text{CeO}_x/\text{Ni}$ catalyst as the erosion depth increased. **These results demonstrated that Ce species were scattered on bulk Ni surface of inverse $4\text{CeO}_x/\text{Ni}$ catalyst, while Ni species were distributed on the CeO_2 surface of conventional $96\text{CeO}_x/\text{Ni}$ catalyst.**”

The corresponding discussion about the **Supplementary Figure 3** were added in the **Supporting Information** as follows:

“*In situ* XPS analysis combined with Ar^+ ion sputtering treatment could give element distribution information along the depth direction with the exposure of the subsurface layer of the catalyst. For the $4\text{CeO}_x/\text{Ni}$ catalyst, the intensity of Ni peak at 852.3 eV (refer to baseline, the same below) increased from 2.0×10^5 to 4.2×10^5 counts after sputtering for 100 seconds (Supplementary Figure 3a). Simultaneously, the corresponding Ce element signal decreased from 1.6×10^4 to 0.7×10^4 counts (calculated based on the characteristic peak at 885.1 eV, Supplementary Figure 3b). **The trend indicated that the relative amount of Ni elements increased from surface to sub-surface of the inverse $4\text{CeO}_x/\text{Ni}$ catalyst after Ar^+ ion sputtering treatment, while the amount of Ce elements corresponding decreased.** In contrast, the Ni signal (peak at 852.3 eV) decreased from 0.7×10^5 to 0.3×10^5 counts (Supplementary Figure 3c) over the $96\text{CeO}_x/\text{Ni}$ catalysts, and the corresponding Ce signal (peak at 882.3 eV) increased from 0.8×10^5 to 1.2×10^5 counts (Supplementary Figure 3d),

which showed that the relative amount of Ce elements increased from surface to sub-surface of the conventional 96CeO_x/Ni catalyst after Ar⁺ ion sputtering treatment, while the amount of Ni elements decreased accordingly.”

As mentioned in above comment, the Ar⁺ ion beam not only sputters some element but also leads to the reduction of many transition-metal oxides, such as CeO₂, TiO₂, and CuO, through the removal of oxygen elements (Wang, G. D., et al. *Appl. Surf. Sci.* **258**, 2057-2061 (2012)). Therefore, the significant chemical valence changes of the Ce species in the 4CeO_x/Ni and 96CeO_x/Ni catalysts after Ar⁺ ion sputtering treatment were observed in **Supplementary Figure 3b and 3d**, respectively.

In order to improve the readability, as suggested, the corresponding discussion was added in the **Supporting Information** as follows:

“It was worth noting that the significant chemical valence changes of the Ce species in the 4CeO_x/Ni and 96CeO_x/Ni catalysts after Ar⁺ ion sputtering treatment could be attributed to removal of oxygen elements by Ar⁺ ion beam sputtering, which led to the reduction of CeO₂.¹

1. Wang, G. D., Kong D. D., Pan Y. H., Pan H. B. & Zhu J. F. Low energy Ar-ion bombardment effects on the ceo2 surface. *Appl. Surf. Sci.* **258**, 2057-2061 (2012).”

Comment 2. *The details of measuring reaction rate were not provided, and it is unclear how the reaction rate was obtained. It would be helpful to know if they were measured at kinetic region by excluding the heat effect and mass transfer effect.*

Response. Thank you for this helpful comment. The reaction as mentioned in **Fig. 1a** was conducted in a continuous flow fixed-bed reactor under controlled reaction conditions (weight hourly space velocity (WHSV) of 0.22 h⁻¹, reaction temperature of 160 °C, 4 MPa H₂, and H₂ flow rate of 50 mL min⁻¹). The catalyst was formed into 40–60 mesh pellets, which were then loaded into the stainless-steel tubular reactor with quartz liner (inner diameter of 10 mm). After the reaction was stable, the kinetic data for the nNi/CeO_x catalysts was obtained with the conversion of tetrahydrofurfuryl alcohol (THFA) kept below 65%. The corresponding reaction rate was calculated using the following equation:

$$r = \frac{\text{molar yield of 1,5-PDO } (\mu\text{mol})}{\text{mass of catalyst (g)} \times \text{reaction time (min)}}$$

Here, the molar yield of 1,5-pentanediol (1,5-PDO) was determined from the conversion of THFA and the selectivity of 1,5-PDO based on three independent measurements as listed in **Supplementary Table 3**. As suggested, the experiment detail was added in the **Method of Main Text** as follows:

“The catalytic activity of the nCeO_x/Ni catalyst was evaluated in a fixed-bed reactor. Typically, 5.4 g of calcinated 4CeO_x/Ni catalyst was formed into 40–60 mesh pellets. The pellets were then loaded into the stainless-steel tubular reactor with quartz liner (inner diameter of 10 mm). Before the reaction, the calcinated catalyst was reduced at 350 °C using a 10 vol % H₂/N₂ flow (40 mL min⁻¹) for 3 hours. After cooling down to the desired temperature, a feed of THFA (solvent free, 0.02 mL min⁻¹) was pumped into the reactor, and the hydrogenolysis reaction was carried out at a temperature of 160 °C under flowing H₂ (4 MPa, 50 mL min⁻¹). During the reaction, the product was collected through a condenser pipe at –10 °C. The collected products were quantitatively analyzed using gas chromatography (GC) equipped with an FFAP chromatographic column and flame ionization detector (Shimadzu GC-2014C). The specific reaction rate was calculated using the following equation:

$$r = \frac{\text{molar yield of 1,5 - PDO } (\mu\text{mol})}{\text{mass of catalyst (g)} \times \text{reaction time (min)}}$$

Here, the molar yield of 1,5-PDO was determined from the conversion of THFA and the selectivity of 1,5-PDO based on three independent measurements.”

Comment 3. *While the interfacial sites were identified as the active sites, a detailed understanding seems to be lacking. For example, the 4CeO₂/Ni sample with four times the CeO₂ amount has only 2 times the reaction rate of that of the 1CeO₂/Ni sample. This suggests that the 1CeO₂/Ni may have higher catalytic efficiency (TOF), provided that these two samples have similar CeO_x size (thus the 4 times of interfacial site number). The authors may want to consider this aspect and have a discussion.*

Response. We appreciate the reviewer for this very insightful comment. Trace or no 1,5-PDO was obtained in the THFA hydrogenolysis over bulk Ni or CeO₂ alone, indicating that the catalytic sites located at the Ce-Ni interface rather than on the CeO₂ or Ni. Recent studies (Chen, A., et al. *Nat. Catal.* **2**, 334-341 (2019), Yan, H., et al. *Nat. Commun.* **10**, 3470 (2019)) revealed that the reaction rate was a function of the interfacial perimeter length. It could be deduced that the reaction rate of interfacial Ce-Ni sites in the THFA hydrogenolysis might also be associated with the interfacial perimeter length. Therefore, we tried to compare the reaction rate of 4CeO_x/Ni and 1CeO_x/Ni catalysts based on the interfacial perimeter length, which was estimated from the size of CeO_x as determined by HRTEM analysis. As shown in **Supplementary Table 2**, the interfacial perimeter length for the 4CeO_x/Ni catalyst was calculated to be 7.5×10⁹ m g⁻¹, which was approximately twice that of the 1CeO_x/Ni catalyst at 4.2×10⁹ m g⁻¹. This could explain why the 4CeO_x/Ni catalyst has a reaction rate normalized by mass of catalyst (29.2 μmol_{1,5-PDO} g⁻¹ min⁻¹) that is about twice as high as that of the 1CeO_x/Ni catalyst (15.4 μmol_{1,5-PDO} g⁻¹ min⁻¹).

The discussion was added in the **Supporting Information** as follows:

“4CeO_x/Ni catalyst has a reaction rate normalized by mass of catalyst (29.2 μmol_{1,5-PDO} g⁻¹ min⁻¹), which was approximately twice as high as that of the 1CeO_x/Ni catalyst (15.4 μmol_{1,5-PDO} g⁻¹ min⁻¹). This higher rate could be attributed to the longer interfacial perimeter length of the 4CeO_x/Ni catalyst, which was 7.5×10⁹ m g⁻¹, about twice that of the 1CeO_x/Ni catalyst at 4.2×10⁹ m g⁻¹.”

The calculation detail was added in the **Supporting Information** as follows:

“**Supplementary Table 2. Physical and chemical characterizations of nCeO_x/Ni catalysts.**

Sample	Ni content ^a (wt %)	Ce content ^a (wt %)	S _{BET} ^b (m ² g ⁻¹)	V _{Pore} ^b (m ³ g ⁻¹)	D _{Pore} ^b (nm)	Interfacial perimeter length ^c (m g ⁻¹)
96CeO _x /Ni	3.8	78.3	92	0.11	4	1.5×10 ⁹
85CeO _x /Ni	14.6	69.5	71	0.09	5	3.1×10 ⁹
4CeO _x /Ni	96.5	2.9	15	0.11	30	7.5×10 ⁹
1CeO _x /Ni	99.1	0.8	2	0.03	73	4.2×10 ⁹

^aActual Ni and Ce contents were determined by ICP-MS. ^bDetermined by N₂ physisorption. ^cThe total length of the nickel-ceria interfacial perimeter was calculated from the size of CeO_x clusters on inverse 1CeO_x/Ni and 4CeO_x/Ni catalysts, and Ni clusters on conventional catalyst, respectively.¹⁵

The perimeter length was calculated using the following equation:

$$Perimeter\ length = \frac{m_{CeO_x/Ni} \times \pi \times d_{CeO_x/Ni}}{\rho_{CeO_x/Ni} \times \frac{1}{2} \times \frac{4}{3} \times \pi \times \left(\frac{d_{CeO_x/Ni}}{2}\right)^3}$$

Here, the model of hemispherical CeO_x or Ni cluster supported on Ni or CeO₂ was used. d_{CeO_x/Ni} represented the particle size of CeO_x or Ni cluster determined from the HRTEM analysis. m_{CeO_x/Ni} and ρ_{CeO_x/Ni} was the loading content of and the density of CeO₂ or Ni, respectively.

15. Yan, H., et al. Construction of stabilized bulk-nano interfaces for highly promoted inverse CeO₂/Cu catalyst. *Nat. Commun.* **10**, 3470 (2019).”

Comment 4. *It is unclear what the actual active sites for the hydrogenolysis of ether are - metal site or the metal oxide site. Metallic metal sites are generally believed to be more effective for the hydrogenolysis, so why does it seem that the oxides are more effective in this work?*

Response. Thank you for the important comments. As suggested, we prepared typical Ni catalysts, including Ni/SiO₂^{R1}, Ni/Al₂O₃^{R2}, and Ni/C^{R3}, where metallic Ni was reported as the active sites for the hydrogenolysis of aryl ethers in the lignin model compounds. However, these Ni catalysts exhibited poor selectivity for 1,5-pentanediol (1,5-PDO) in the selective ring-opening hydrogenolysis of tetrahydrofurfuryl alcohol (THFA). Instead, methyltetrahydrofuran (MTHF) was obtained with selectivity over 50% (**Table R1**). These results proved that the metallic Ni sites was not the actual active sites for the hydrogenolysis of etheric C–O bond of THFA. Furthermore, as shown in **Fig. 1a**, trace or no 1,5-PDO was obtained in the THFA hydrogenolysis over bulk Ni or CeO₂ alone, indicating that the interfacial Ce–Ni sites rather than the CeO₂ or Ni was responsible for the hydrogenolysis of etheric C–O. The further experiments and DFT calculation also demonstrated that the high catalytic activity of the 4CeO_x/Ni catalyst in the THFA hydrogenolysis was associated with the interfacial Ce–Ni sites delivered by the inverse oxide/metal configuration, where the Ce species selectively adsorbed etheric oxygen of THFA (**Fig. 3**) and the Ni species promoted the formation and adsorption of H^{δ-} species (**Fig. 4**), both of them cooperatively promoted the hydrogenolysis of THFA.

Table R1. Selective hydrogenolysis of THFA over various catalysts in a batch reactor.^[a]

Catalyst	Conversion (%)	Selectivity (%)				Ref.
		1,5-PDO	1-butanol	1-pentanol	MTHF	
4CeO _x /Ni	100	98	<1	<1	<1	This work
Ni/SiO ₂	6	<1	<1	<1	66	R1
Ni/Al ₂ O ₃	15	<1	<1	<1	48	R2
Ni/C	10	<1	<1	<1	67	R3

[a] Reaction conditions: catalyst, 0.25 g; THFA, 0.5 g; temperature, 160 °C; ethanol, 5 mL; 4.0 MPa H₂; 24 h. MTHF = methyltetrahydrofuran

- R1. He, J., Zhao C. & Lercher J. A. Ni-catalyzed cleavage of aryl ethers in the aqueous phase. *J. Am. Chem. Soc.* **134**, 20768-20775 (2012).
- R2. Qi, L., *et al.* Unraveling the dynamic network in the reactions of an alkyl aryl ether catalyzed by Ni/γ-Al₂O₃ in 2-propanol. *J. Am. Chem. Soc.* **141**, 17370-17381 (2019).
- R3. Song, Q., *et al.* Lignin depolymerization (LDP) in alcohol over nickel-based catalysts via a fragmentation–hydrogenolysis process. *Energy Environ. Sci.* **6**, 994-1007 (2013).

The above results were added in the **Supporting Information** as follows:

“Supplementary Table 4. Selective hydrogenolysis of THFA over various catalysts in a batch reactor.”^[a]

Entry ^[b]	Catalyst	Conversion (%)	Selectivity (%)			
			1,5-PDO	1-butanol	1-pentanol	MTHF
1	IM-CeO _x /Ni	19	95	<1	<1	<1
2	IM-ReO _x /Ni	2	63	1	2	22
3	IM-MoO _x /Ni	7	68	2	10	20
4	IM-VO _x /Ni	5	62	2	2	8
5	IM-WO _x /Ni	2	69	<1	11	9
6 ^[c]	4CeO _x /Ni	35	98	<1	<1	<1
7 ^[c, d]	4CeO _x /Ni	100	98	<1	<1	<1
8 ^[d]	Ni/SiO ₂ ^[e]	6	<1	<1	<1	66
9 ^[d]	Ni/Al ₂ O ₃ ^[f]	15	<1	<1	<1	48
10 ^[d]	Ni/C ^[g]	10	<1	<1	<1	67

[a] Reaction conditions: catalyst, 0.25 g; THFA, 0.5 g; temperature, 160 °C; ethanol, 5 mL; 4.0 MPa H₂; 10 h. [b] The catalysts of IM-M⁺O_x/Ni (M⁺=Re, Mo, V, W, molar ratio of M⁺/Ni = 1/80) catalysts in entry 1–5 were prepared by impregnation method (IM). [c] 4CeO_x/Ni catalyst was prepared by coprecipitation method (CP). [d] 24 h. [e] Ref. 16. [f] Ref. 17. [g] Ref. 18. MTHF = methyltetrahydrofuran

16. He, J., Zhao C. & Lercher J. A. Ni-catalyzed cleavage of aryl ethers in the aqueous phase. *J. Am. Chem. Soc.* **134**, 20768-20775 (2012).
17. Qi, L., et al. Unraveling the dynamic network in the reactions of an alkyl aryl ether catalyzed by Ni/γ-Al₂O₃ in 2-propanol. *J. Am. Chem. Soc.* **141**, 17370-17381 (2019).
18. Song, Q., et al. Lignin depolymerization (LDP) in alcohol over nickel-based catalysts via a fragmentation–hydrogenolysis process. *Energy Environ. Sci.* **6**, 994-1007 (2013).”

Comment 5. *When comparing the production rate of 1,5-PDO, the content of Ni or Ce should also be considered. It is important to determine the critical active sites for the hydrogenolysis of ether - metal site or the metal oxide site.*

Response. Thank you for this comment. As mentioned above, the actual catalytic sites are supposed to locate at the interfacial Ce-Ni sites rather than on the metal or metal oxide sites, since trace or no 1,5-PDO was obtained in the THFA hydrogenolysis over bulk Ni or CeO₂ alone. Therefore, the reaction rate normalized by the Ce mass or Ni mass might be inappropriate for understanding the difference in catalytic performance between the inverse and conventional interfacial Ce-Ni sites. Even so, to provide a more complete picture of the relative activity of each catalyst, the reaction

rate of THFA hydrogenolysis over $n\text{CeO}_x/\text{Ni}$ catalysts normalized by their Ce mass and Ni mass was provided in the **Supplementary Fig. 4c and 4d of Supporting Information**. The representative inverse $4\text{CeO}_x/\text{Ni}$ catalysts also exhibited a higher 1,5-PDO production rate than the representative conventional $96\text{CeO}_x/\text{Ni}$ catalysts.

Supplementary Figure 4. The reaction rate of THFA hydrogenolysis over $n\text{CeO}_x/\text{Ni}$ catalyst normalized by their (a) interfacial perimeter length, (b) surface areas, (c) Ce mass, and (d) Ni mass.”

Comment 6. *In the EPR results, the shift of the g factor may not directly reflect the electron density on Ni.*

Response. Thank you for this important reminder. It is reported that the shift of the g value could be ascribed to the partial transfer of unpaired electron for the Ni–O–Ce catalyst, contributed by the strong interaction between Ni and Ce ions (Wang, J. Q., et al. *Catal. Lett.* **140**, 38-48 (2010)). Therefore, the g-value of Ni species shifted from 2.39 in the conventional $96\text{CeO}_x/\text{Ni}$ catalysts to 2.16 in the inverse $4\text{CeO}_x/\text{Ni}$ catalysts could be attributed to strong interaction between Ni and Ce species. The discussion in the **Main Text** was revised as follows:

“...The lower reduction temperature of interfacial Ce species on the $4\text{CeO}_x/\text{Ni}$ and $1\text{CeO}_x/\text{Ni}$ catalysts indicated the stronger interaction between interfacial Ni and Ce on inverse catalyst,⁴⁷ which could be further demonstrated by *quasi in situ* electron paramagnetic resonance (EPR) spectroscopy, where the g-value of Ni species shifted from 2.39 in the conventional $96\text{CeO}_x/\text{Ni}$ catalysts to 2.16 in the inverse $4\text{CeO}_x/\text{Ni}$ catalysts (Supplementary Fig. 8d).^{42, 48}”

48. Wang, J. Q., et al. Effects of Ni-doping of ceria-based materials on their micro-structures and dynamic oxygen storage and release behaviors. *Catal. Lett.* **140**, 38-48 (2010).

Comment 7. *The electron density of interfacial Ni species and Ce species in $4\text{CeO}_x/\text{Ni}$ is lower*

than in $96\text{CeO}_x/\text{Ni}$. The definition and distinction of Ni at the interface should be elucidated.

Response. Thank you very much for this helpful reminder. The mentioned interfacial Ni species in the work was defined as the Ni adjacent to the Ce atom, which was labelled as $\text{Ni}(-\text{Ce})_{\text{interface}}$ in this manuscript. For readers' exact understanding, the definition was added in the **Main Text**, and the corresponding caption in **Fig. 2c** and **Fig. 4c** were revised as follows:

“Ce in the bulk CeO_2 (labeled as Ce_{bulk}), interfacial Ce without adjacent oxygen vacancy (labeled as $\text{Ce}(-\text{O})_{\text{interface}}$), and interfacial Ce associated with an oxygen vacancy (labeled as $\text{Ce}(-\text{V}_\text{O})_{\text{interface}}$). The corresponding Ni species were denoted as Ni_{bulk} and $\text{Ni}(-\text{Ce})_{\text{interface}}$, where the latter was defined as the interfacial Ni adjacent to Ce.”

Fig. 2 The structural analysis of $n\text{CeO}_x/\text{Ni}$ catalysts. c Bader charge analysis of Ce and Ni species within inverse $\text{Ce}_{10}\text{O}_{19}/\text{Ni}(111)$ and conventional $\text{Ni}_{10}/\text{CeO}_2(111)$ catalysts, respectively.

Fig. 4. Quantitative analysis of hydrogen species on catalyst surface. c Bader charge analysis of H species on $\text{Ce}_{10}\text{O}_{19}/\text{Ni}(111)$ and $\text{Ni}_{10}/\text{CeO}_2(111)$.

Comment 8. While the writing is quite good, there are still a few poorly constructed sentences.

For example, “While the strong diffraction peaks of 96CeO_x/Ni and 85CeO_x/Ni catalysts were assigned to the characteristic fluorite CeO₂ phase (PDF No: 65-5923),” needs to be revised.

Response. Thank you for this kind reminder. We have revised this sentence in the **Main Text** as follows:

“Using X-ray powder diffraction (XRD), the remarkable Ni crystal phase (PDF No: 65-2865) was observed in the 4CeO_x/Ni and 1CeO_x/Ni catalysts (Supplementary Fig. 1), and the negligible CeO₂ diffraction peaks could be ascribed to the low content of Ce and the high dispersion of CeO_x cluster.³³ In contrast, the strong diffraction peaks observed in the 96CeO_x/Ni and 85CeO_x/Ni catalysts indicated the presence of the characteristic fluorite CeO₂ phase (PDF No: 65-5923).”

Reviewer #3

Comment. *Li et al introduce via this study the use of inverse Ce/Ni based catalyst with controlling its initial composition. The authors claims that the 4CeO_x/Ni catalyst is the most active for the upgrading of the oxygen containing biomass and plastic wastes and prob reactions for this study. The current research showcases the potential in adjusting the interfacial electronic structure of inverse catalysts help to show better activity than traditional metal-oxide supported catalysts for these reactions. However, it necessitates further elucidation and examination of additional aspects before consideration for publication in Nature Communications.*

Response. We thank the reviewer very much for the positive comment and kind encouragement.

Comment 1. *Some observation was presented without an explanation in several key areas. For example, there was no clarification on why cerium (Ce) peaks were absent in the X-Ray Diffraction (XRD) for the 4CeO_x/Ni, as illustrated in figures S1, with only nickel (Ni) characteristic peaks being observed.*

Response. Thank you for this insightful comment. As shown in **Figure S1**, the characteristic diffraction peaks of CeO₂ were not observed in the XRD patterns of the 4CeO_x/Ni and 1CeO_x/Ni catalysts, which was attributed to the low content of Ce (4 wt.% and 1 wt.%, respectively) and the high dispersion of CeO_x cluster with small particle size of 2~3 nm observed by TEM images and EDS mapping (**Supplementary Figure 2**). Moreover, the increased crystal structure disorder of the CeO_x cluster on the 4CeO_x/Ni and 1CeO_x/Ni catalysts, indicated by the negligible F_{2g} bands observed in the Raman spectra (**Fig. 2a**), could also be associated with the absence of the characteristic CeO₂ diffraction peaks. To clarify the absence of CeO₂ peaks for the 4CeO_x/Ni and 1CeO_x/Ni catalysts, the discussion in the **Main Text** was revised as follows:

“Using X-ray powder diffraction (XRD), the remarkable Ni crystal phase (PDF No: 65-2865) was observed in the 4CeO_x/Ni and 1CeO_x/Ni catalysts (Supplementary Fig. 1), **and the negligible CeO₂ diffraction peaks could be ascribed to the low content of Ce and the high dispersion of CeO_x cluster.**³³ In contrast, the strong diffraction peaks observed in the 96CeO_x/Ni and 85CeO_x/Ni catalysts indicated the presence of the characteristic fluorite CeO₂ phase (PDF No: 65-5923).”

33. Yan, H., et al. Construction of stabilized bulk-nano interfaces for highly promoted inverse CeO₂/Cu catalyst. *Nat. Commun.* **10**, 3470 (2019).

Comment 2. *Additionally, the X-ray Photoelectron Spectroscopy (XPS) data require further elucidation concerning the observed trends, particularly why both Ni and Ce signals diminish in the 4CeO_x/Ni catalyst, a trend not echoed in the 96CeO_x/Ni catalyst.*

Response. Thank you very much for this insightful comment. We guessed the X-ray Photoelectron Spectroscopy (XPS) data you are referring to was about the XPS analysis combined with Ar⁺ ion sputtering treatment. As shown in **Supplementary Figure 3a and b**, for the inverse 4CeO_x/Ni catalyst, the intensity of Ni peak at 852.3 eV (refer to baseline) increased from 2.0×10⁵ to 4.2×10⁵ counts after sputtering for 100 seconds. Simultaneously, the corresponding Ce element signal decreased from 1.6×10⁴ to 0.7×10⁴ counts (calculated based on the characteristic peak at 885.1 eV). The trend indicated that the relative amount of Ni elements increased from surface to sub-surface of the inverse 4CeO_x/Ni catalyst after Ar⁺ ion sputtering treatment, while the amount of Ce elements corresponding decreased. In contrast, according to the results in **Supplementary Figure 3c and d**, the Ni signal (peak at 852.3 eV) decreased from 0.7×10⁵ to 0.3×10⁵ counts over the 96CeO_x/Ni catalysts, and the corresponding Ce signal (peak at 882.3 eV) increased from 0.8×10⁵ to 1.2×10⁵ counts, which showed that the relative amount of Ce elements from surface and sub-surface of the conventional 96CeO_x/Ni catalyst increased after Ar⁺ ion sputtering treatment, while the amount of Ni elements corresponding decreased. The above results disclosed that Ce elements were scattered on bulk Ni surface of inverse 4CeO_x/Ni catalyst, while Ni elements were distributed on the CeO₂ surface of conventional 96CeO_x/Ni catalyst.

The corresponding discussions about the **Supplementary Figure 3** were added in the **Supporting Information** as follows:

“*In situ* XPS analysis combined with Ar⁺ ion sputtering treatment could give element distribution information along the depth direction with the exposure of the subsurface layer of the catalyst. For the 4CeO_x/Ni catalyst, the intensity of Ni peak at 852.3 eV (refer to baseline, the same below) increased from 2.0×10⁵ to 4.2×10⁵ counts after sputtering for 100 seconds (Supplementary Figure 3a). Simultaneously, the corresponding Ce element signal decreased from 1.6×10⁴ to 0.7×10⁴ counts (calculated based on the characteristic peak at 885.1 eV, Supplementary Figure 3b). The trend indicated that the relative amount of Ni elements increased from surface to sub-surface of the inverse 4CeO_x/Ni catalyst after Ar⁺ ion sputtering treatment, while the amount of Ce elements corresponding decreased. In contrast, the Ni signal (peak at 852.3 eV) decreased from 0.7×10⁵ to 0.3×10⁵ counts (Supplementary Figure 3c) over the 96CeO_x/Ni catalysts, and the corresponding Ce signal (peak at 882.3 eV) increased from 0.8×10⁵ to 1.2×10⁵ counts (Supplementary Figure 3d), which showed that the relative amount of Ce elements increased from surface to sub-surface of the conventional 96CeO_x/Ni catalyst after Ar⁺ ion sputtering treatment, while the amount of Ni elements decreased accordingly.”

Comment 3. *There's also a need to address the dimensions of the CeO_x clusters within the 4CeO_x/Ni and 1CeO_x/Ni to compare these with the size of Ni in the conventional samples. It is important to evaluate how these observed sizes align with modeled structure by Density Functional Theory (DFT) modeling, potentially necessitating a discussion on size distribution*

and particle counts.

Response. Thank you for this important suggestion. As shown in **Supplementary Figure 2**, the EDS mapping indicates that the Ce species was well-dispersed over the 4CeO_x/Ni and 1CeO_x/Ni catalysts, and the Ni was highly dispersed on the 96CeO_x/Ni and 85CeO_x/Ni catalysts. Due to the low contrast between CeO_x and Ni, it is difficult to distinguish between Ce and Ni and measure the particle size of CeO_x using the HAADF-STEM. Therefore, the size of CeO_x clusters and Ni particle was measured using HTREM images. The results show that the CeO_x clusters on the 4CeO_x/Ni and 1CeO_x/Ni catalyst is approximately 2~3 nm in size. The particle size of Ni on the conventional 96CeO_x/Ni catalyst was about 6 nm, increasing to approximately 8 nm for the 85CeO_x/Ni catalyst. Based on the above experimental results and the reported works, we selected the model of Ce₁₀O₁₉ cluster supported on Ni(111) as a representative of inverse catalyst (Li, Y., et al, *ACS Catal.* **10**, 3164-3174 (2020)), and the model of CeO₂(111) to support Ni₁₀ cluster as a representative of conventional catalysts (Xie, J., et al. *ACS Catal.* **13**, 9577-9587 (2023)), respectively.

The corresponding description and discussion were added in the **Main Text** as follows:

“Their great contrast in compositions suggested the different structural configurations of Ni and CeO₂ in the nCeO_x/Ni catalysts, whose geometrical structure was studied by high-resolution transmission electron microscopy (HRTEM), high-angle annular dark field scanning transmission electron microscopy (HAADF-STEM), and energy dispersive X-ray spectroscopy (EDS). As shown in **Supplementary Fig. 2a**, a representative inverse oxide/metal configuration of well-dispersed nano CeO_x clusters (size of approximately 3 nm) supported on the Ni surface was observed over the 4CeO_x/Ni catalyst. In contrast, the 96CeO_x/Ni catalyst has a typical metal/oxide configuration, where the Ni particles (size of approximately 6 nm) were supported on the bulk CeO₂ (Supplementary Fig. 2b).”

As suggested, the corresponding reference was added in the **Main Text** as follows:

“Therefore, the structure of interface Ce-Ni was explored by using theoretical calculations with the models of Ce₁₀O₁₉/Ni(111) and Ni₁₀/CeO₂(111) representing the inverse and conventional catalyst, respectively.^{49,50}

50. Xie, J., et al. Hydrogenolysis of lignin model compounds on Ni nanoparticles surrounding the oxygen vacancy of CeO₂. *ACS Catal.* **13**, 9577-9587 (2023).”

Supplementary Figure 2. HRTEM images, HAADF-STEM images, and the corresponding EDS mapping of (a) $4\text{CeO}_x/\text{Ni}$, (b) $96\text{CeO}_x/\text{Ni}$, (c) $1\text{CeO}_x/\text{Ni}$, and (d) $85\text{CeO}_x/\text{Ni}$ catalysts.

Comment 4. *Moreover, the reason behind the increase in oxygen vacancies in the $4\text{CeO}_x/\text{Ni}$ following tetrahydrofuran (THF) adsorption.*

Response. Thank you for this important comment. We conducted the XPS analysis for the inverse $4\text{CeO}_x/\text{Ni}$ catalyst before and after tetrahydrofurfuryl alcohol (THFA) adsorption to reveal the adsorption mode of THFA on the catalyst. As shown in **Supplementary Figure 14a**, no remarkable change in ratio of $\text{Ce}^{3+}/(\text{Ce}^{3+}+\text{Ce}^{4+})$ was observed on inverse $4\text{CeO}_x/\text{Ni}$ catalyst (from 67% to 65%), which indicated the oxygen vacancy did not remarkably increase or decrease after THFA adsorption, since the formation of oxygen vacancy was closely associated with the reduction of Ce^{4+} into Ce^{3+} . The increase in the relative content of O_{II} species from 1.6 to 2.9 was observed on the $4\text{CeO}_x/\text{Ni}$ catalyst (**Supplementary Figure 14a**). In the XPS spectra of O 1s, the oxygen species of O_{II} belongs to the surface-adsorbed oxygen species, typically associated with the oxygen vacancy (Chen, A., et al. *Nat. Catal.* **2**, 334-341 (2019), Zheng, X., et al. *ACS Catal.* **10**, 3968-3983 (2020)). Therefore, it could be deduced that the increase of O_{II} species was attributed to the adsorption of oxygen atom of THFA at the $\text{Ce}(-\text{V}_{\text{O}})_{\text{interface}}$ site adjacent to the oxygen vacancy. The corresponding discussion in the **Main Text** was revised as follows:

“The relative content of oxygen vacancy (determined by the ratio of $\text{Ce}^{3+}/(\text{Ce}^{3+}+\text{Ce}^{4+})$) and the surface-adsorbed O species associated with oxygen vacancy (denoted as O_{II} , calculated by its corresponding area ratio to lattice oxygen of O_{I}) were monitored, respectively.^{35,58} As shown in Supplementary Fig. 14, the relative content of O_{II} species increased from 1.6 to 2.9 with no remarkable change of oxygen vacancy after THFA adsorption over the inverse $4\text{CeO}_x/\text{Ni}$ catalyst. By contrast, the THFA adsorbed on conventional $96\text{CeO}_x/\text{Ni}$ catalyst led to the decrease in O_{II} content from 0.6 to 0.5 with the diminishment of oxygen vacancy content ($\text{Ce}^{3+}/(\text{Ce}^{3+}+\text{Ce}^{4+})$) ratio

from 41% to 34%). These results manifested the adsorption mode of THFA over inverse 4CeO_x/Ni catalyst via selective adsorption of etheric oxygen at the Ce(-V_O)_{interface} site adjacent to oxygen vacancy, while both the adsorption of etheric oxygen at the Ni(-Ce)_{interface} site and deprotonation of hydroxyl at the oxygen vacancy occurred simultaneously over conventional 96CeO_x/Ni catalyst.”

35. Chen, A., et al. Structure of the catalytically active copper-ceria interfacial perimeter. *Nat. Catal.* **2**, 334-341 (2019).

58. Zheng, X., et al. Highly efficient porous Fe_xCe_{1-x}O_{2-δ} with three-dimensional hierarchical nanoflower morphology for H₂S-selective oxidation. *ACS Catal.* **10**, 3968-3983 (2020).

The corresponding description in the **Supporting Information** was also revised.

“To study the adsorption sites for the oxygenic group of THFA, the oxygen species on the catalyst were analyzed by quasi in situ XPS. The reduced catalyst was first impregnated in THFA and then vacuumed overnight to remove the physical adsorbed THFA for the following XPS analysis. As shown in Supplementary Figure 14a,b, the XPS spectra of O 1s could be deconvoluted into three peaks at 529.1 eV, 530.9 eV, and 532.4 eV, which are assigned to lattice oxygen of CeO₂ (O_I), the surface-adsorbed oxygen species associated with oxygen vacancy (O_{II}), and adsorbed water (O_{III}), respectively.^{2,3} The relative content of O_{II} was calculated by its corresponding area ratio to O_I, since the lattice oxygen species (O_I) was stable before and after THFA adsorption.”

2. Chen, A., et al. Structure of the catalytically active copper-ceria interfacial perimeter. *Nat. Catal.* **2**, 334-341 (2019).

3. Zheng, X., et al. Highly efficient porous Fe_xCe_{1-x}O_{2-δ} with three-dimensional hierarchical nanoflower morphology for H₂S-selective oxidation. *ACS Catal.* **10**, 3968-3983 (2020).

Comment 5. *Similarly, an exploration is needed into why the 1CeO_x/Ni sample exhibits a lower surface area and how this characteristic influences the catalytic activity.*

Response. Thank you for this suggestion. As shown in **Figure R1**, the scanning electron microscopy (SEM) images indicated that the Ni particle size of 4CeO_x/Ni (approximately 43 nm) is smaller than that of 1CeO_x/Ni (70 nm), which led to the decrease in the specific surface area from 15 m² g⁻¹ for 4CeO_x/Ni to 2 m² g⁻¹ for 1CeO_x/Ni.

Generally, the catalyst possessed larger surface area afforded higher catalytic reaction rate. However, the 1CeO_x/Ni and 4CeO_x/Ni catalysts exhibited a higher 1,5-PDO production rate than the 85CeO_x/Ni and 96CeO_x/Ni catalysts, even though the specific surface area of 1CeO_x/Ni (2 m² g⁻¹) and 4CeO_x/Ni (15 m² g⁻¹) were smaller than those of 85CeO_x/Ni (71 m² g⁻¹) and 96CeO_x/Ni (92 m²

g⁻¹). These results indicated the relationship between the catalytic activity of nCeO_x/Ni catalysts in the THFA hydrogenolysis and their surface area is not significantly close.

Figure R1. SEM images of (a) 4CeO_x/Ni and (b) 1CeO_x/Ni catalysts.

Comment 6. *It is suggested that Energy Dispersive X-ray Spectroscopy (EDS) mapping be incorporated to monitor the dispersion of Ce and Ni atoms on the catalyst surfaces, which could provide valuable insights into the spatial distribution and interaction of these elements.*

Response. Thank you for this helpful suggestion. As suggested, the high-angle annular dark field scanning transmission electron microscopy (HAADF-STEM) images, and the energy dispersive X-ray spectroscopy (EDS) mapping of nCeO_x/Ni catalysts (**Supplementary Figure 2**) were provided in the **Supporting Information**. The EDS mapping reveals that Ce was well-dispersed on Ni over the inverse 4CeO_x/Ni and 1CeO_x/Ni catalysts, and Ni was uniformly dispersed on CeO₂ over the 96CeO_x/Ni and 85CeO_x/Ni catalysts. The corresponding description and discussion were added in the **Main Text** as follows:

“Their great contrast in compositions suggested the different structural configurations of Ni and CeO₂ in the nCeO_x/Ni catalysts, whose geometrical structure was studied by high-resolution transmission electron microscopy (HRTEM), high-angle annular dark field scanning transmission electron microscopy (HAADF-STEM), and energy dispersive X-ray spectroscopy (EDS). As shown in **Supplementary Fig. 2a**, a representative inverse oxide/metal configuration of well-dispersed nano CeO_x clusters (size of approximately 3 nm) supported on the Ni surface was observed over the 4CeO_x/Ni catalyst. In contrast, the 96CeO_x/Ni catalyst possessed the typical metal/oxide configuration, where the Ni particles (size of approximately 6 nm) were supported on the bulk CeO₂ (**Supplementary Fig. 2b**).”

The **Supplementary Figure 2** and the corresponding description were added in the **Supporting Information** as follows:

Supplementary Figure 2. HRTEM images, HAADF-STEM images, and the corresponding EDS mapping of (a) $4\text{CeO}_x/\text{Ni}$, (b) $96\text{CeO}_x/\text{Ni}$, (c) $1\text{CeO}_x/\text{Ni}$, and (d) $85\text{CeO}_x/\text{Ni}$ catalysts.

“The HRTEM images show that the CeO_x cluster, approximately 2~3 nm in size, were supported on Ni over the inverse $4\text{CeO}_x/\text{Ni}$ and $1\text{CeO}_x/\text{Ni}$ catalysts. The particle size of Ni on the conventional $96\text{CeO}_x/\text{Ni}$ catalyst was 6 nm, increasing to approximately 8 nm for the $85\text{CeO}_x/\text{Ni}$ catalyst. The EDS mapping reveals that Ce is well-dispersed on the Ni over the inverse $4\text{CeO}_x/\text{Ni}$ and $1\text{CeO}_x/\text{Ni}$ catalysts, while Ni was uniformly dispersed on CeO_2 over the $96\text{CeO}_x/\text{Ni}$ and $85\text{CeO}_x/\text{Ni}$ catalysts.”

Comment 7. *The authors assigned the enhanced of activity to the strong interfacial electronic interaction between the oxide and metal, which promoted the cleavage of etheric C–O bond. What about increase the ratio of the Ceria size would that increase the interface and thus the activity.*

Response. Thank you very much for this interesting suggestion.

As reported in the literatures (Chen, A., et al. *Nat. Catal.* **2**, 334-341 (2019), Yan, H., et al. *Nat. Commun.* **10**, 3470 (2019)), the reaction rate has been found to be a function of the interfacial perimeter length. Therefore, it could be deduced that the reaction rate of interfacial Ce-Ni sites in the THFA hydrogenolysis might also be associated with the interfacial perimeter length, which could be estimated from the size of CeO_x as determined by HRTEM analysis. Therefore, it could be deduced that increasing the number of the CeO_x cluster could increase the interfacial perimeter length, which led to the improvement of catalytic activity. However, preparing high-dispersed CeO_x with small size on metal is challenging due to the facile formation of large CeO_x cluster through aggregation of small CeO_x cluster with increasing Ce loading, which led to the decrease in perimeter length and thus the activity. Therefore, the appropriate Ce loading is the key to preparing the optimal Ce-Ni catalyst.

Comment 8. *Would other ratios like 10 or 20- CeO_x/Ni still be a good candidate for the reaction as well?*

Response. Thank you for this suggestion. As suggested, we prepared the 20-CeO_x/Ni and 10-CeO_x/Ni catalysts using the same coprecipitation method, with Ni/Ce mole ratios of 20:1 and 10:1, respectively. The corresponding hydrogenolysis of THFA was conducted in a fixed-bed reactor, and the results were listed in **Table R2**. The 20-CeO_x/Ni and 10-CeO_x/Ni catalysts, albeit inferior to the 4CeO_x/Ni catalyst, exhibited acceptable hydrogenolysis activity with THFA conversion of 52% and 49%, respectively.

Table R2. Selective hydrogenolysis of THFA over nCeO_x/Ni catalysts in a fixed-bed reactor^[a]

Catalyst	Conversion (%)	Selectivity (%)			
		1,5-PDO	1-butanol	1-pentanol	MTHF
20-CeO _x /Ni (Ni:Ce=20:1)	53	97	2	<1	<1
	51	98	1	<1	<1
	52	96	3	<1	<1
10-CeO _x /Ni (Ni:Ce=10:1)	48	97	1	<1	<1
	49	97	2	<1	<1
	50	98	1	<1	<1
4CeO _x /Ni	63	98	2	<1	<1
	63	99	1	<1	<1
	64	97	2	<1	<1

[a] Reaction conditions: calcinated catalyst: 5.4 g; THFA (solvent free): 0.02 mL min⁻¹; temperature: 160 °C, H₂ pressure: 4.0 MPa, gas-flow rate: 50 mL min⁻¹. The data were collected from three independent measurements. MTHF = methyltetrahydrofuran.

Reviewer #4

Comment. *In this study, the authors synthesized and characterized a catalyst comprised of ceria supported on nickel metal and employed it for carbon-oxygen bond hydrogenolysis of tetrahydrofurfural alcohol (THFA) and other model compounds. The “inverse catalyst”, denoted 4CeOx/Ni, was compared to the traditional ceria-supported nickel analogue, denoted 96CeOx/Ni, and found to have improved activity and selectivity. The authors performed a variety of complementary characterization techniques to support the theory that the 4CeOx/Ni has sufficient electronic structure to activate C-O bonds via oxygen vacancies and hydrogen on metallic sites, including FTIR, XPS, UV-Raman, UV-Vis, EPR, and H₂-TPD-MS. The catalysts were also characterized with TEM and ICP-MS. The “quasi in-situ” studies with XPS and EPR involved reduction of the catalysts followed by air-free transfer to the instrument, where the catalyst was saturated with THFA then vacuumed to only leave behind physisorbed material. Calculations based upon Density Functional Theory (DFT) were also used to compare the energetics of C-O bond cleavage over the different structures, suggesting the 4CeOx/Ni provides favorable adsorption at electron-rich Ce and Ni interfacial sites. The authors performed stability tests and used the catalyst for several other biomass-relevant model compounds including 5-hydroxymethyl furfural (HMF), veratrylglycerol-β-guaiacyl ether (VG), and polyethylene glycol (PEG). The implementation of the inverse supported catalyst for carbon-oxygen bond hydrogenolysis is timely and interesting from the perspective of bond activation. But overall, the application for biomass and plastic is very preliminary. Furthermore, several concerns regarding the activity comparison, stability experiments, quasi in-situ characterization, and model compounds should be addressed. If addressed, this paper might be suitable for publication in Nature Communications.*

Response. We appreciate the reviewer very much for the positive comment and encouragement.

Comment 1. *For the literature presented in Supplementary Table 1, are the authors attributing the deactivation of Nickel-based catalysts to leaching, or general deactivation as evidenced by conversion loss? Is leaching a concern for the ceria-nickel based catalyst in this study?*

Response. Thank you for this comment. There are various reasons for the deactivation of nickel-based catalysts. For example, for the Ni/ZSM-5 catalyst (**Supplementary Table 1, entry 8**), the decrease in the conversion of THFA from 93% to 74% was attributed to the aggregation of Ni nanoparticles (Soghrati, E., et al. *Appl. Catal., B* **235**, 130-142 (2018)). While, for the Ni-WO_x/SiO₂ catalyst (**Supplementary Table 1, entry 9**), the conversion of THFA declined from 39% to 29% after 2 cycles, which was ascribed to the leaching of W species (Soghrati, E., et al. *ChemCatChem* **10**, 4652-4664 (2018)). In addition, the leaching of Ni (Wang, Z., et al. *Catal. Today* **402**, 79-87 (2022)) could lead to the decrease in the conversion of THFA from 50% to 43% over the NiPr/Al₂O₃

catalyst (**Supplementary Table 1, entry 11**). In this work, the 4CeO_x/Ni catalyst exhibited high stability during the 200-h reaction in the fixed-bed reactor, without observed leaching of Ce and Ni species, as determined by ICP examination of reaction solution. Moreover, the agglomeration of metallic Ni species was negligible according to the XRD results (**Supplementary Fig. 7a**).

Comment 2. *The rates in Figure 1 are on a mass of catalyst basis- Did the authors compare selectivity and activity on a nickel basis? The mass basis might over-estimate activity.*

Response. Thank you for this suggestion. As shown in **Fig. 1a**, trace or no 1,5-PDO was obtained in the THFA hydrogenolysis over bulk Ni or CeO₂ alone, indicating that the actual catalytic sites located at the Ce-Ni interface rather than on the CeO₂ or Ni. Hence, the reaction rate based the Ni mass might not be appropriate for understanding the difference in catalytic performance between the inverse and conventional interfacial Ce-Ni sites. As suggested, the reaction rate of THFA hydrogenolysis over nCeO_x/Ni catalysts was normalized by Ni mass (**Supplementary Fig. 4d**). The results demonstrated that the representative inverse 4CeO_x/Ni catalyst exhibited superior catalytic activity compared to the representative conventional 96CeO_x/Ni catalyst.

As reported in the literatures (Chen, A., et al. *Nat. Catal.* **2**, 334-341 (2019), Yan, H., et al. *Nat. Commun.* **10**, 3470 (2019)), the reaction rate was found to be a function of the interfacial perimeter length. It could be deduced that the reaction rate of interfacial Ce-Ni sites in the THFA hydrogenolysis might also be associated with the interfacial perimeter length. Therefore, we tried to compare the reaction rate of 4CeO_x/Ni and 96CeO_x/Ni catalysts based on the interfacial perimeter length, which was estimated from the size of CeO_x and Ni cluster as determined by HRTEM analysis (**Supplementary Table 2**). As shown in **Supplementary Fig. 4a**, the inverse 4CeO_x/Ni catalyst exhibited a higher production rate of 1,5-PDO (3.9 μmol_{1,5-PDO} m⁻¹ min⁻¹) than that of conventional 96CeO_x/Ni catalyst (0.53 μmol_{1,5-PDO} m⁻¹ min⁻¹), which indicated that the trend of production rate normalized by interfacial interface length was the same with that on a mass of catalyst basis. In the reported the literatures, the production rate based on the mass of catalyst was widely used for evaluating the catalytic activity of inverse catalyst (Yan, H., et al. *Nat. Commun.* **10**, 3470 (2019), Xu, K., et al. *Nat. Commun.* **13**, 2443 (2022)). Therefore, the production rate normalized by the mass of catalyst is appropriate for comparing the catalytic activity of inverse and conventional catalysts.

Supplementary Figure 4. The reaction rate of THFA hydrogenolysis over $n\text{CeO}_x/\text{Ni}$ catalyst normalized by their (a) interfacial perimeter length, (b) surface areas, (c) Ce mass, and (d) Ni mass.

Comment 3. *The methods for the quasi in-situ XPS and EPR measurements are unclear. Are these measurements done at room temperature? What are the key differences between the conditions of the measurement and the actual experiment? What are the limitations of this measurement?*

Response. Thank you for this helpful comment. The *quasi in-situ* XPS measurement was conducted for analyzing the valence state of Ni, Ce, and O elements in the $n\text{CeO}_x/\text{Ni}$ catalysts. Firstly, the catalyst was reduced in a glass tube, which was then sealed and transferred into a glove box without exposure to air. Subsequently, the sample was loaded on a sealed specimen stage. Lastly, the specimen stage was transferred into the XPS analysis chamber. Similarly, the *quasi in-situ* EPR sample was encapsulated in a glass capillary inside a glove box. All of the operations were carried out at room temperature inside a glove box. The key difference between the condition of the measurements and the actual experiment was that the actual experiment was conducted in high-pressure H_2 (4 MPa), whereas the XPS and EPR measurements were performed in an ultra-high vacuum and in N_2 at atmospheric pressure, respectively. The XPS experiment under high-pressure is still challenging, while it is delightful that the developed near-ambient pressure XPS (NAP-XPS) for the *in-situ* experiments, albeit the pressure in the scale of mbar, has attracted increased attention (Lian, X., *J. Phys. Chem. Lett.* **13**, 8264-8277 (2022)). As suggested, the method details for the *quasi in-situ* XPS and EPR measurements were revised in the **Method of Main Text** as follows:

“*Quasi in situ* XPS experiments were performed on a Thermo Scientific NEXSA Instrument. Before the XPS measurements, the catalyst was reduced in a glass tube, which was then sealed and transferred into a glove box without exposure to air. Subsequently, the sample was loaded on a

sealed specimen stage and transferred into the XPS analysis chamber. All of the operations were carried out at room temperature inside a glove box. Ultraviolet Raman spectra were collected on a Labram HR Evolution (HORIBA) with a semiconductor laser (325 nm, 1 mW). The diffuse reflection ultraviolet-visible spectra were collected on a UV-vis spectrophotometer (Shimadzu, UV-2700). *Quasi in situ* EPR experiments were carried out on a JEOL RESONANCE JES-X320 spectrometer operating at X-band frequency ($\nu \approx 9.15$ GHz) at 25 °C. Before the EPR measurements, the catalyst powder was reduced in a glass tube, which was then sealed and transferred into a glove box without exposure to air. Subsequently, the sample was sealed in a glass capillary under a N₂ atmosphere. All of the operations were performed at room temperature inside a glove box.”

Comment 4. *For stability measurements, the authors ran the 4CeOX/Ni catalyst for 200 hours for tetrahydrofurfuryl alcohol (THFA) conversion into 1,5-pentanediol (1,5-PDO). The authors used 13 gram of catalyst, and ran at 100% conversion. Given the large catalyst loading, and complete conversion, it is difficult to tell if the catalyst is truly stable or if the catalyst loading is so high that no loss in conversion is observed. It is recommended that the authors use a lower loading and operate at a lower conversion and lower spacetimes to characterize the stability of the catalyst.*

Response. Thank you for this helpful suggestion. As suggested, the controlled reaction with about half conversion of THFA was conducted in a continuous flow fixed-bed reactor under the similar reaction condition, except for the increase in the feed rate of THFA from 0.02 mL min⁻¹ to 0.04 mL min⁻¹ (calcinated catalyst: 13 g; reaction temperature of 160 °C, 4 MPa H₂, and H₂ flow rate of 50 mL min⁻¹). As shown in **Supplementary Fig. 6**, the catalyst also worked stably for over 200 h, affording >97% selectivity of 1,5-PDO under the THFA conversion of about 55%. These results indicated the catalyst was stable for the hydrogenolysis of THFA under the reaction conditions. As suggested, the result was added to the **Supporting Information** as **Supplementary Figure 6**, and the corresponding discussion was added in the **Main Text** as follows:

“Notably, the inverse 4CeO_x/Ni catalyst demonstrated good stability for 200 h in a fixed-bed reactor without remarkable decrease in the THFA conversion or 1,5-PDO selectivity at half and full conversion of THFA (Fig. 1b, Supplementary Fig. 6), which could be associated with the inhibited agglomeration of metallic Ni species (Supplementary Fig. 7a)³⁸ and the enhanced stability in the Ce³⁺ species of CeO_x cluster (Supplementary Fig. 7b)⁵.”

Supplementary Figure 5. Catalyst stability experiment in a fixed-bed reactor over a 4CeO_x/Ni catalyst. Before the reaction, the calcinated catalyst was pre-reduced in a H₂/N₂ flow. Reaction conditions: calcinated catalyst: 13 g; THFA (solvent-free): 0.04 mL min⁻¹; 160 °C; 4.0 MPa H₂; gas-flow rate: 50 mL min⁻¹.

Comment 5. *For the model compound studies (HMF, VG, PEG), did the authors reduce the catalyst first and perform air-free transfer into the batch reactor vessel?*

Response. Thank you for this comment. For the hydrogenolysis reaction of compounds, such as THFA, HMF, VG, and PEG, in a batch reactor, the catalyst was firstly reduced at 350 °C for 3 hours in a 10 vol % H₂/N₂ flow and subsequently passivated in a 5 vol% O₂/N₂ flow at 30 °C for 5 min. The catalyst was weighted and transferred into the batch reactor in the air. The experimental detail was added in the **Method** of **Main Text** as follows:

“For the reaction in a batch reactor, the catalyst was initially reduced at 350 °C for 3 hours in a 10 vol % H₂/N₂ flow (50 mL min⁻¹) and subsequently passivated in a 5 vol% O₂/N₂ flow (40 mL min⁻¹) at 30 °C for 5 mins. Prior to the reaction, the autoclave was purged three times and then pressured with H₂ gas.”

Comment 6. *The authors compared the performance of the 4CeO_x/Ni catalyst to the 96CeO_x/Ni for the HMF, VG, and PEG substrates, with significantly lower yields over the 96CeO_x/Ni. Given the 24-fold increase in nickel, in the reverse catalyst, how would the yields compare if the catalysts were normalized by Ni loading?*

Response. Thank you for this insightful suggestion. As shown in **Table R3**, the low yields of the specific products from the transformation of HMF, VG, and PEG over the 96CeO_x/Ni were attributed to the low selectivity. Therefore, it could be deduced that the yields of the corresponding products would not increase even more 96CeO_x/Ni catalyst was used.

Table R3. Selective hydrogenolysis of representative lignocellulose derivatives over 4CeO_x/Ni and 96CeO_x/Ni catalysts in a batch reactor.

Substrate	Catalyst	Conversion (%)	Selectivity (%)		
 (HMF) ^a	4CeO _x /Ni	100	 5	 91	
	96CeO _x /Ni	100	83	8	
 (VG) ^b	4CeO _x /Ni	100	 83	 4	 2
	96CeO _x /Ni	100	26	45	5
 (PEG-200) ^c	4CeO _x /Ni	100	 85	 2	
	96CeO _x /Ni	100	12	80	

[a] catalyst, 0.25 g; HMF, 0.5 g; temperature, 160 °C; ethanol, 5 mL; 4.0 MPa H₂; 24 h. [b] catalyst, 0.12 g; HMF, 0.25 g; temperature, 150 °C; ethanol, 5 mL; 6.0 MPa H₂; 36 h. [c] catalyst, 0.4 g; PEG-200, 0.5 g; temperature, 160 °C; ethanol, 5 mL; 4.0 MPa H₂; 48 h. HMF = 5-hydroxymethyl furfural. VG = veratrylglycerol-β-guaiacyl ether. PEG = polyethylene glycol.

Comment 7. *For PEG experiments, was there evidence that the solvent (THF) was converted?*

Response. Thank you for this comment. Using gas chromatography equipped with mass spectrometry (GC-MS), the possible hydrogenolysis products, such as 1-butanol and n-butane, were not detected in the reaction of PEG, suggesting that no or only trace conversion of THF was obtained under the reaction conditions for the hydrogenolysis of PEG.

Comment 8. *Does PEG depolymerization occur over this catalyst in the absence of hydrogen? Would this catalyst be effective and/or practical for glycolysis/methanolysis/hydrolysis? Technically ethanol is not the monomer of PEG, it is ethylene glycol. What is the proposed mechanism here for ethanol formation?*

Response. Thank you for this insightful question. When the reaction of PEG was performed in N₂ atmosphere, the possible depolymerization products such as ethylene glycol or ethanol were not observed, suggesting hydrogen is indispensable for the hydrogenolysis of PEG over the 4CeO_x/Ni catalyst. To avoid interference from possible depolymerization products, the methanolysis and hydrolysis of PEG were carried out in methanol and water rather than ethylene glycol, respectively.

Trace methanolysis product of 1,2-dimethoxyethane and hydrolysis product of ethylene glycol were detected in the reaction solution, indicating the catalyst was not effective for the methanolysis and hydrolysis of PEG under the hydrogenolysis reaction conditions. As illustrated in below **Scheme R2**, ethylene glycol could be obtained from PEG in the presence of water via hydrolysis pathway, while the hydrogenation depolymerization of PEG catalyzed by inverse 4CeO_x/Ni catalyst proceeded through the hydrogenolysis of etheric C-O bond in the presence of H₂, which produced the ethanol rather than ethylene glycol.

Scheme R2. Hydrolysis and hydrogenolysis transformation of PEG in the presence of water and H₂, respectively.

Comment 9. *Have the authors tried using polyethylene terephthalate (PET) or other polyesters for depolymerization? To make the claim of the utility of this catalyst beyond model compounds, the authors would have to demonstrate its applicability with additional polymeric materials.*

Response. Thank you very much for this helpful suggestion. As suggested, the transformation of polyethylene terephthalate (PET) and poly(bisphenol A-co-epichlorohydrin) glycidyl end-capped (PBAE) were carried out under the similar reaction condition with the hydrogenolysis of THFA. The inverse 4CeO_x/Ni catalyst hydrogenation converted PET into dimethyl cyclohexane dicarboxylate with a higher yield of 80% than that of 8% yield over the 96CeO_x/Ni catalyst. 4CeO_x/Ni catalyst also demonstrated outstanding activity (53% yield) in the depolymerization of PBAE to produce propane-2,2-diylidicyclohexane and 4-(2-cyclohexylpropan-2-yl)phenol via selective hydrogenolysis of etheric C-O bond. These results show the great potential of such inverse 4CeO_x/Ni catalyst in the recycling of plastics waste. These results were added in **Supplementary Table 5 and Fig. 5**, and the corresponding experimental details and discussions were revised as follows:

“The reaction of 0.5 g of polyethylene terephthalate (PET, white granular, intrinsic viscosity of 0.82

dL/g) was carried out in methanol under 170 °C (0.4 g catalyst, 4.0 MPa H₂). The conversion of poly(bisphenol A-co-epichlorohydrin) glycidyl end-capped (PBAE, Mn ≈ 1075) was carried out in methanol (0.4 g catalyst, 170 °C, 4.0 MPa H₂).”

“...4CeO_x/Ni also depolymerized the poly(bisphenol A-co-epichlorohydrin) glycidyl end-capped (PBAE) via selective hydrogenolysis of etheric C–O bond, achieving a 53% yield of propane-2,2-diyldicyclohexane and 4-(2-cyclohexylpropan-2-yl)phenol. In addition, the 4CeO_x/Ni catalyst demonstrated high catalytic activity in the hydrogenation conversion of polyethylene terephthalate (PET) to dimethyl cyclohexane dicarboxylate in 80% yield. These results show the great potential of such inverse 4CeO_x/Ni catalyst in the recycle of plastics waste.”

Fig. 5. Transformation of representative lignocellulose derivatives and polyether/ester over inverse 4CeO_x/Ni catalyst.

Supplementary Table 5. Selective hydrogenation of representative lignocellulose derivatives and polyether/ester over 4CeO_x/Ni and 96CeO_x/Ni catalysts in a batch reactor.

Substrate	Catalyst	Conversion (%)	Selectivity (%)		
 (HMF) ^a	4CeO _x /Ni	100			91
	96CeO _x /Ni	100	83	8	
 (VG) ^b	4CeO _x /Ni	100				96CeO _x /Ni	100	26	45	5

(PEG-200) ^c	4CeO _x /Ni	100	85	2	
	96CeO _x /Ni	100	12	80	
(PET) ^d	4CeO _x /Ni	100	80	12	3
	96CeO _x /Ni	90	9	trace	89
(PBAE) ^e	4CeO _x /Ni	100	33	20	29
	96CeO _x /Ni	70	trace	trace	10

[a] catalyst, 0.25 g; HMF, 0.5 g; temperature, 160 °C; ethanol, 5 mL; 4.0 MPa H₂; 24 h. [b] catalyst, 0.12 g; HMF, 0.25 g; temperature, 150 °C; ethanol, 5 mL; 6.0 MPa H₂; 36 h. [c] catalyst, 0.4 g; PEG-200, 0.5 g; temperature, 160 °C; ethanol, 5 mL; 4.0 MPa H₂; 48 h. [d] catalyst, 0.4 g; PET, 0.5 g; temperature, 170 °C; methanol, 10 mL; 4.0 MPa H₂; 24 h. [e] catalyst, 0.25 g; PBAE, 0.5 g; temperature, 170 °C; methanol, 10 mL; 4.0 MPa H₂; 24 h. HMF = 5-hydroxymethyl furfural. VG = veratrylglycerol-β-guaiacyl ether. PEG = polyethylene glycol. PET = polyethylene terephthalate. PBAE = poly(bisphenol A-co-epichlorohydrin) glycidyl end-capped.”

REVIEWERS' COMMENTS

Reviewer #1 (Remarks to the Author):

The authors have provided thorough and satisfactory responses to the comments. I believe the revisions have improved the manuscript. I have no further comments.

Reviewer #2 (Remarks to the Author):

After reviewing the authors' response and the revised manuscript, I believe they have satisfactorily addressed all of my questions and concerns. Therefore, I would like to recommend acceptance at this time.

Reviewer #3 (Remarks to the Author):

The authors have effectively addressed the raised comments, resulting in a more coherent and clarified manuscript. No further comments from my end.

Reviewer #4 (Remarks to the Author):

In this revised manuscript, the authors addressed many of the reviewer's comments through additional experimentation and discussion. One of the key improvements was the improved rigor of the normalization experiments to demonstrate that the interfacial sites are required for high selectivity towards 1,5-PDO. The authors also added clarification as to the techniques for XPS and added more stability data at lower conversions. The authors also added additional experiments with PET to showcase the potential for polyester deconstruction. Overall, the manuscript is greatly improved, showcasing a unique catalyst architecture for sustainable transformations, and is suitable for publication in Nature Communications.

Responses to Reviewers' comments

Reviewer #1

Comment. *The authors have provided thorough and satisfactory responses to the comments. I believe the revisions have improved the manuscript. I have no further comments.*

Response. Thank you very much for the kind comment and encouragement.

Reviewer #2

Comment. *After reviewing the authors' response and the revised manuscript, I believe they have satisfactorily addressed all of my questions and concerns. Therefore, I would like to recommend acceptance at this time.*

Response. We thank the reviewer very much for the positive comment and encouragement.

Reviewer #3

Comment. *The authors have effectively addressed the raised comments, resulting in a more coherent and clarified manuscript. No further comments from my end.*

Response. Thank the reviewer very much for the positive comment and encouragement.

Reviewer #4

Comment. *In this revised manuscript, the authors addressed many of the reviewer's comments through additional experimentation and discussion. One of the key improvements was the improved rigor of the normalization experiments to demonstrate that the interfacial sites are required for high selectivity towards 1,5-PDO. The authors also added clarification as to the techniques for XPS and added more stability data at lower conversions. The authors also added additional experiments with PET to showcase the potential for polyester deconstruction. Overall, the manuscript is greatly improved, showcasing a unique catalyst architecture for sustainable transformations, and is suitable for publication in Nature Communications.*

Response. We thank the reviewer very much for the positive comment and kind encouragement.